# FIXING MODEL-FITTING: COMPRESSING GUIDANCE FOR BETTER DIFFUSION SAMPLING

## ABSTRACT

Model-fitting occurs when samples are overly adjusted to satisfy with the guidance model rather than the true conditions, often leading to poor outcomes. The root cause of this problem is the consecutiveness of guidance timesteps throughout the diffusion sampling process. In this work, We quantify this effect and show that breaking the consecutiveness of standard guidance alleviates the problem. Based on this insight, our method, Compress Guidance, distributes a small number of guidance steps across the full sampling process, yielding substantial improvements in image quality and diversity while cutting guidance cost by over 80%. Experiments on both label-conditional image generation and text-to-image generation, across multiple datasets and models, confirm that Compress Guidance consistently surpasses baselines in image quality with significantly lower computational overhead.

## 1 INTRODUCTION

Guidance is mainly divided into classifier-free guidance in Ho & Salimans (2022) and classifier guidance in Dhariwal & Nichol (2021). Although both of these methods significantly improve the performance of the diffusion samples Dhariwal & Nichol (2021); Ho & Salimans (2022); Bansal et al. (2023); Liu et al. (2023); Epstein et al. (2023), they both suffer from high computation cost. For classifier guidance, the act of gradients calculation backwards through a classifier is costly. On the other hand, forwarding through a diffusion model twice at every timestep also costs significant computation in classifier-free guidance.

This work challenges the necessity of the current complex process based on several key observations. First, we find that the guidance loss is predominantly active during the early stages of the sampling process, when the image lacks a well-defined structure. As the model progresses and shifts its focus to refining image details, the guidance loss tends to approach zero. Additionally, when evaluating intermediate samples with an additional classifier not used for guidance, we observe that the loss from this external classifier does not decrease in the same way as it does for the guidance-specific classifier. This suggests that the generated samples are tailored to fit the features of the guiding classifier rather than producing generalized features applicable to different classifiers. We define this issue as *model-fitting*, where the generated image pixels are optimized to satisfy the guiding classifier's criteria rather than generalizing to the intended conditions. The problem is validated by three pieces of evidence in section 4.1.

These observations prompt us to question whether guidance is necessary at every timestep and how reducing the frequency of guidance could enhance generative quality. In Section 4.2, we further explore the properties of guidance in ensuring sample quality. Based on this analysis, we propose a simple yet effective method called Compress Guidance (CompG), which mitigates the issue by reducing the number of time steps that invoke gradient calculation. This approach not only improves sample quality but also significantly accelerates the overall process as shown in Fig.1. In most parts of the works, we utilize classifier guidance as the main object for observations due to the explicit loss given by the classifier. However, the methods can be applied to classifier-free guidance as well.

Concurrent works have explored relevant ideas. Wang et al. (2024) shows that early guidance can cause conflicts and degrade outputs, while IntervalGuidance Kynkäänniemi et al. (2025) reports the best results at mid-range noise timesteps. However, as demonstrated in Section 4.2, these insights are not always correct. Moreover, the cost reduction in IntervalGuidance arises only as a byproduct

A cat is on a couch

| Stable Diffusion | CLIP score: 30
running time: 5.5 sec/image | **Ours** | CLIP score: **31**
running time: **3.75** sec/image |

Figure 1: *Stable Diffusion with classifier-free guidance. The left figure is the vanilla classifier-free guidance with application on all 50 timesteps. Our proposed Compress Guidance method is the right figure, where we only apply guidance on 8 out of 50 steps. Our methods are superior to classifier-free guidance regarding image quality, quantitative performance, and efficiency. The efficiency is calculated by sampling 30000 images with 1 GPU. More comparison is in Appendix H (Figure 13, 14, 15, 16)*

of avoiding low/high noise conflicts; it lacks an explicit mechanism for controlling guidance cost. In contrast, our method directly regulates guidance steps through a defined equation, improving efficiency without sacrificing performance. To the best of our knowledge, we are the first to explicitly reduce guidance cost without relying on additional distillation training, and we do so by principled analysis of model-fitting challenges. Furthermore, our plug-and-play approach is compatible with any model. Throughout most of our study, we observe on classifier guidance because the classifier provides explicit loss signals (unlike classifier-free guidance). Nevertheless, our observations extend naturally to classifier-free guidance, as discussed in Section 4.4.

Overall, the contributions of our works are three-fold: **(1)** Explore and quantify the model-fitting problem in guidance and the redundant computation resulting from current guidance methods. **(2)** Propose a simple but effective method to contain the model-fitting problem and improve computational time. **(3)** Extensive analysis and experimental results for different datasets and generative tasks on both classifier and classifier-free guidance perspectives.

## 2 RELATED WORK

Diffusion Generative Models (DGMs) Ho et al. (2020); Song et al. (2020b); Vahdat et al. (2021); Song & Ermon (2020); Lipman et al. (2022) have recently become one of the most popular generative models in many tasks such as image editingKawar et al. (2023); Huang et al. (2024), text-to-image sampling Rombach et al. (2022); Podell et al. (2023); Ramesh et al. (2022) or image/videos generation Ho et al. (2022); Blattmann et al. (2023). Guidance is often utilized to improve the performance of DGMs Dhariwal & Nichol (2021); Ho & Salimans (2022); Bansal et al. (2023); Liu et al. (2023); Epstein et al. (2023); Wang et al. (2024); Karras et al. (2023; 2022). Besides improving the performance, the guidance also offers a trade-off between image quality and diversity Dhariwal & Nichol (2021); Ho & Salimans (2022); Ma et al. (2023), which helps users tune their sampling process up to their expectations. Gradient views of guidance are also well explored in the literature. Zheng et al. (2022) explores the gradient vanishing problem in classifier guidance, while Dinh et al. (2023a) examines conflicts in guidance sampling, and Dinh et al. (2023b) investigates guidance uncertainty during sampling. Our work introduces a new perspective on guidance—model-fitting—drawing an analogy to the overfitting problem in neural network training.

Although guidance is beneficial in many forms, it faces severe serious drawbacks in running time. For classifier guidance, the running time is around 80% higher compared to the original diffusion model

sampling time due to the evaluation of gradients at every sampling step. In contrast, classifier-free guidance requires the process to forward to the expensive diffusion model twice at every timestep. Previous works on improving the running time of DGMs involve the reduction of sampling steps Song et al. (2020a); Zhang & Chen (2022); Song et al. (2023) and latent-based diffusion models Rombach et al. (2022); Peebles & Xie (2023). Recently, the research community has focused on distilling from a large number of timesteps to a smaller number of timesteps Salimans & Ho (2022); Sauer et al. (2023); Li et al. (2024); Yin et al. (2024b;a); Song & Dhariwal (2023); Wang et al. (2025); Meng et al. (2023); Heek et al. (2024); Yan et al. (2024); Luhman & Luhman (2021); Ren et al. (2025); Zhou et al. (2024) or reducing the architectures of diffusion models Li et al. (2024); Tang et al. (2023); Zhang et al. (2024). However, most of these works mainly solve the problem of the expensive diffusion samplings, not the cost resulted by guidance.

Prior work Wang et al. (2024); Kynkäänniemi et al. (2025) shows that early guidance can harm generation in conditional diffusion models due to conflicts with conditional inputs. However, this does not apply to unconditional models, where early guidance is essential for quality. These studies also overlook the high computational cost of guidance. In contrast, our work addresses both quality and efficiency, generalizing model-fitting issues to all guided diffusion and proposing a plug-and-play solution to reduce guidance overhead without sacrificing performance.

## 3 BACKGROUND

**Diffusion Models** Ho et al. (2020) have the form of: $p_\theta := p(\mathbf{x}_T) \prod_{t=1}^{T} p_\theta(\mathbf{x}_{t-1}|\mathbf{x}_t)$ where $p_\theta(\mathbf{x}_{t-1}|\mathbf{x}_t) := \mathcal{N}(\mathbf{x}_{t-1}; \mu_\theta(x_t, t), \Sigma_\theta(x_t, t))$ supporting the reverse process from $\mathbf{x}_T$ to $\mathbf{x}_0$. This process is denoising process where starting from the $\mathbf{x}_T \sim \mathcal{N}(\mathbf{x}_T; 0, \mathbf{I})$ to gradually move to $\mathbf{x}_0 \sim q(\mathbf{x}_0)$. This process is trained to be matched with the forward diffusion process $q(\mathbf{x}_{1:T}|\mathbf{x}_0) := \prod_{t=1}^{T} q(\mathbf{x}_t|\mathbf{x}_{t-1})$ given $q(\mathbf{x}_t|\mathbf{x}_{t-1})$ as $q(\mathbf{x}_t|\mathbf{x}_{t-1}) := \mathcal{N}(\mathbf{x}_t; \sqrt{1-\beta_t}\mathbf{x}_{t-1}, \beta\mathbf{I})$ or we can write the conditional distribution of $\mathbf{x}_t$ given $\mathbf{x}_0$ as below:

$$q(\mathbf{x}_t|\mathbf{x}_0) := \mathcal{N}(\mathbf{x}_t; \sqrt{\bar{\alpha}_t}\mathbf{x}_0, (1-\bar{\alpha}_t)\mathbf{I}) \tag{1}$$

$\beta_t$ is the fixed variance scheduled before the process starts, Ho et al. (2020) denotes $\alpha_t := 1 - \beta_t$ and $\bar{\alpha}_t := \prod_{s=1}^{t} \alpha_s$ used in Eq.1. We have the $\mathbf{x}_{t-1}$ conditioned on $\mathbf{x}_0$ and $\mathbf{x}_t$ as:

$$q(\mathbf{x}_{t-1}|\mathbf{x}_t, \mathbf{x}_0) = \mathcal{N}(\mathbf{x}_{t-1}; \tilde{\boldsymbol{\mu}}_t(\mathbf{x}_t, \mathbf{x}_0), \tilde{\beta}_t\mathbf{I}) \tag{2}$$

where $\tilde{\boldsymbol{\mu}}_t(\mathbf{x}_t, \mathbf{x}_0) := \frac{\sqrt{\bar{\alpha}_{t-1}}\beta_t}{1-\bar{\alpha}_t}\mathbf{x}_0 + \frac{\sqrt{\alpha_t}(1-\bar{\alpha}_{t-1})}{1-\bar{\alpha}_t})\mathbf{x}_t$ and $\tilde{B}_t := \frac{1-\bar{\alpha}_{t-1}}{1-\bar{\alpha}_t}\beta_t$. To train the diffusion model, the lower bound loss is utilized as below:

$$\mathbb{E}[-\log p_\theta(\mathbf{x}_0)] \leq \mathbb{E}_q[-\log p(\mathbf{x}_T) - \Sigma_{t\geq 1}\log \frac{p_\theta(\mathbf{x}_{t-1}|\mathbf{x}_t)}{q(\mathbf{x}_t|\mathbf{x}_{t-1})}] \tag{3}$$

Rewrite Eq. 3 as $\mathbb{E}_q[D_{KL}(q(\mathbf{x}_T|\mathbf{x}_0)||p(\mathbf{x}_T)) + \sum_{t>1} D_{KL}(q(\mathbf{x}_{t-1}|\mathbf{x}_t, \mathbf{x}_0)||p_\theta(\mathbf{x}_{t-1}|\mathbf{x}_t)) - \log p_\theta(\mathbf{x}_0|\mathbf{x}_1)]$ The training process actually optimize the $\sum_{t>1} D_{KL}(q(\mathbf{x}_{t-1}|\mathbf{x}_t, \mathbf{x}_0)||p_\theta(\mathbf{x}_{t-1}|\mathbf{x}_t))$ where the diffusion model try to match the distribution of $\mathbf{x}_{t-1}$ by using only $\mathbf{x}_t$. There are several implementations for optimising the 3. However, the $\theta$ as parameters of the noise predictor $\epsilon_\theta(\mathbf{x}_t, t)$ is the most popular choice. After the $\theta$ are trained using Eq. 3, we have the sampling equation:

$$\mathbf{x}_{t-1} = \frac{1}{\sqrt{\alpha_t}}(\mathbf{x}_t - \frac{1-\alpha_t}{\sqrt{1-\bar{\alpha}_t}}\epsilon_\theta(\mathbf{x}_t, t)) + \sigma_t\mathbf{z} \tag{4}$$

**Guidance** in the Diffusion model offers conditional information and image quality enhancement. Given a classifier $p_\phi(y|\mathbf{x}_t)$ that match with the labels distribution conditioned on images $\mathbf{x}_t$, we have the sampling equation with guidance as:

$$\mathbf{x}_{t-1} \sim \mathcal{N}(\mu_t + s\sigma_t^2\nabla_{\mathbf{x}_t}\log p_\phi(y|\mathbf{x}_t), \sigma_t) \tag{5}$$

with $s$ is the guidance scale. Besides the classifier guidance as Eq.5, Ho & Salimans (2022) proposes another version named classifier-free guidance. This guidance method does not base the information on a classifier. Instead, the guidance depends on the conditional information from a conditional diffusion model. The sampling equation has the form:

$$\mathbf{x}_{t-1} \sim \mathcal{N}(\tilde{\boldsymbol{\mu}}_t(\mathbf{x}_t, \frac{\mathbf{x}_t - \sqrt{1-\bar{\alpha}}\tilde{\epsilon}_t}{\sqrt{\bar{\alpha}_t}}), \sigma_t) \tag{6}$$

given $\tilde{\epsilon} = (1+w)\epsilon_\theta(\mathbf{x}_t, c) - w\epsilon_\theta(\mathbf{x}_t)$ with $w$ is the guidance scale.

## 4 MODEL-FITTING IN GUIDANCE

We begin by modelling the sampling equation as two distinct optimization objectives, illustrating that the sampling process functions as a form of "training", where parameters $\mathbf{x}_t$ are optimized over $T$ timesteps. We then analyze the "training" of $\mathbf{x}_t$ in light of these objectives, highlighting the model-fitting problem that arises in the current guidance-driven sampling process. From Eq.4:

$$\mathbf{x}_{t-1} = \frac{(1-\alpha_t)\sqrt{\bar{\alpha}_{t-1}}}{1-\bar{\alpha}_t} \frac{\mathbf{x}_t - \sqrt{1-\bar{\alpha}_t}\epsilon_\theta(\mathbf{x}_t, t)}{\sqrt{\bar{\alpha}_t}} + \frac{(1-\bar{\alpha}_{t-1})\sqrt{\alpha_t}}{1-\bar{\alpha}_t}\mathbf{x}_t + \sigma_t z \tag{7}$$

**Distribution matching objective**: Assuming that $\epsilon_\theta(\mathbf{x}_t, t)$ is learned perfectly to match random noise $\epsilon$ at timestep $t$, we have $\frac{\mathbf{x}_t - \sqrt{1-\bar{\alpha}_t}\epsilon_\theta(\mathbf{x}_t, t)}{\sqrt{\bar{\alpha}_t}} = \mathbf{x}_0$ is the exact prediction of $\mathbf{x}_0$ at timestep $t$ according to Eq.1. Denoting $\tilde{\mathbf{x}}_0$ is the prediction of $\mathbf{x}_0$ at timestep $t$, we can re-write the equation as bellow:

$$\mathbf{x}_{t-1} = \frac{(1-\alpha_t)\sqrt{\bar{\alpha}_{t-1}}}{1-\bar{\alpha}_t}\tilde{\mathbf{x}}_0 + \frac{(1-\bar{\alpha}_{t-1})\sqrt{\alpha_t}}{1-\bar{\alpha}_t}\mathbf{x}_t + \sigma_t z \tag{8}$$

This equation 8 can be derived from $q(\mathbf{x}_{t-1}|\mathbf{x}_t, \mathbf{x}_0)$ in Eq. 2 with parameterized trick for Gaussian Distribution. Thus, the first aim of the sampling process is to match the distribution $q(\mathbf{x}_{t-1}|\mathbf{x}_t, \mathbf{x}_0)$. Nevertheless, the Eq.8 is based on the assumption that $\tilde{\mathbf{x}}_0 \sim \mathbf{x}_0$, which often does not hold when $t \to T$. Given $\tilde{\mathbf{x}}_0 = \frac{\mathbf{x}_t - \sqrt{1-\bar{\alpha}_t}\epsilon_\theta(\mathbf{x}_t, t)}{\sqrt{\bar{\alpha}_t}}$, this formulation is rooted from $\tilde{\mathbf{x}}_0 \sim \mathcal{N}(\frac{1}{\sqrt{\bar{\alpha}}}\mathbf{x}_t; \frac{\bar{\alpha}-1}{\bar{\alpha}}\mathbf{I})$ with assumption that $\epsilon_\theta(\mathbf{x}_t, t) \sim \epsilon$. However, $\epsilon_\theta(\mathbf{x}_t, t)$ is trained to minimize $D_{KL}[q(\mathbf{x}_{t-1}|\mathbf{x}_t, \mathbf{x}_0)||p_\theta(\mathbf{x}_{t-1}|\mathbf{x}_t)]$ as in Ho et al. (2020) which actually causes a significantly distorted information if $\epsilon_\theta(\mathbf{x}_t, t)$ is utilized to sample $\tilde{\mathbf{x}}_0$ from $\mathbf{x}_t$ if $t \to T$. A smaller $t$ would result in a better prediction of $\mathbf{x}_0$ and with $t = 0$, we have $\bar{\alpha} = 1$ resulting in $\tilde{\mathbf{x}}_0 = \mathbf{x}_t$.

**Theorem 1.** *Assuming $\epsilon_\theta$ is trained to converge with noise prediction error magnitude at a timestep $t$ is approximate $\Delta$, the sampling process $\mathbf{x}_{t-1} \sim q(\mathbf{x}_{t-1}|\mathbf{x}_t, \tilde{\mathbf{x}}_0)$ from $T$ to $0$ is equivalent to the minimization of $||\mathbf{x}_0 - \tilde{\mathbf{x}}_0||$ .wrt. $\mathbf{x}_t$.*

Full proof is shown in Appendix F. If we consider $\mathbf{x}_t$ of the Eq.8 as the set of optimization parameters, the sampling process will have the objective $\min_{\mathbf{x}_t} ||\mathbf{x}_0 - \tilde{\mathbf{x}}_0||$. The Eq.8 turns into:

$$\mathbf{x}_{t-1} = \mathbf{x}_t - \underbrace{(\frac{\sqrt{\alpha_t}-1}{\sqrt{\alpha_t}}\mathbf{x}_t + \frac{1-\alpha_t}{\sqrt{1-\bar{\alpha}_t}\sqrt{\alpha_t}}\epsilon_\theta(\mathbf{x}_t, t) - \sigma_t \mathbf{z})}_{\gamma_1 \nabla_{\mathbf{x}_t} ||\mathbf{x}_0 - \tilde{\mathbf{x}}_0||} \tag{9}$$

Eq.9 turns the sampling process into a stochastic gradient descent process where the $\mathbf{x}_t$ is the parameter of the model at the timestep $t$, the updated direction into $\mathbf{x}_t$ aims to satisfy $\min_{\mathbf{x}_t} ||\mathbf{x}_0 - \tilde{\mathbf{x}}_0||$. We denote gradient $\nabla_{\mathbf{x}_t} ||\mathbf{x}_0 - \tilde{\mathbf{x}}_0||$ as denoising gradient $\nabla_{\mathbf{x}_t} d$.

**Classification objective**: From Eq.5, we have the term $s\sigma_t^2 \nabla_{\mathbf{x}_t} \log p_\phi(y|\mathbf{x}_t)$ is added to the sampling equation for guidance. This term can be written in full form as $s\sigma_t^2 \nabla_{\mathbf{x}_t}(q(y)\log q(y) - q(y)\log p_\phi(y|\mathbf{x}_t))$ which is equivalent to $-s\sigma_t^2 \nabla D_{KL}[q(y)||p_\phi(\hat{y}|\mathbf{x}_t)]$. Combine Eq.9 with guidance information in Eq.5, we have:

$$\mathbf{x}_{t-1} = \mathbf{x}_t - \underbrace{(\frac{\sqrt{\alpha_t}-1}{\sqrt{\alpha_t}}\mathbf{x}_t + \frac{1-\alpha_t}{\sqrt{1-\bar{\alpha}_t}\sqrt{\alpha_t}}\epsilon_\theta(\mathbf{x}_t, t) - \sigma_t \mathbf{z})}_{\gamma_1 \nabla_{\mathbf{x}_t} d} - \underbrace{(-s\sigma_t^2 \nabla_{\mathbf{x}_t} \log p_\phi(y|\mathbf{x}_t))}_{\gamma_2 \nabla_{\mathbf{x}_t} D_{KL}[q(y)||p_\phi(\hat{y}|\mathbf{x}_t)]} \tag{10}$$

As a result, the process of updating $\mathbf{x}_t$ to $\mathbf{x}_{t-1}$ is a "training" step to optimize to objective functions $||\mathbf{x}_0 - \tilde{\mathbf{x}}_0||$ and $D_{KL}[q(y)||p_\phi(\hat{y}|\mathbf{x}_t)]$ with two gradients respecting to $\mathbf{x}_t$ as Eq.10. Since this is similar to the training process, it is expected to face some problems in training deep neural networks. In this work, the problem of model fitting is detected by observing the losses given by the classification objective during the sampling process. For convenience, from later on, $\nabla$ is denoted for $\nabla_{\mathbf{x}_t}$. Although classifier-free guidance does not have explicit loss function like classifier guidance, the observation on classifier guidance still can be applied to classifier-free guidance.

### 4.1 MODEL-FITTING

Based on the optimization problem from the sampling process in the previous section, we first define *on-sampling loss* and *off-sampling loss* for observation.

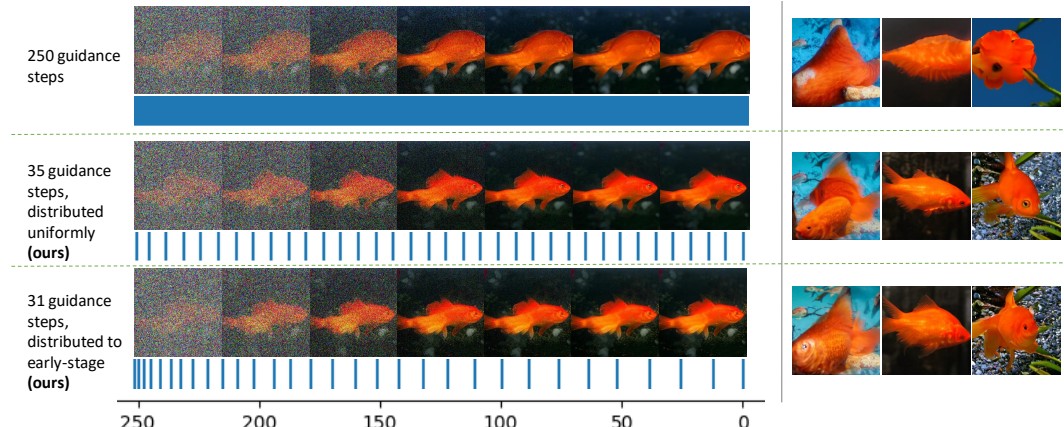

Figure 2: *ImageNet256x256. The top row is the vanilla guidance (obtained by ADM-G Dhariwal & Nichol (2021)), where all the timesteps got the guidance information. The second and third rows are our proposed method, which only applies 35 time steps. The second row distributes the timesteps uniformly, while the third row distributes the timesteps toward the early stage of the sampling process. The Compress Guidance performs significantly better than the original guidance method. One blue stick means one guidance step.*

**Definition 1.** *On-sampling loss/accuracy refers to the loss or accuracy evaluated on the generated samples $\mathbf{x}_t$ at timestep $t$ during the diffusion sampling process, which consists of $T$ timesteps. This loss is defined as $-\log p_\phi(\hat{y}|\mathbf{x}_t)$ by the classifier parameters $\phi$ that provide guidance throughout the sampling process.*

**Definition 2.** *Off-sampling loss/accuracy refers to the loss or accuracy evaluated on the generated samples $\mathbf{x}_t$ at timestep $t$ during the diffusion sampling process, which consists of $T$ timesteps. This loss is defined as $-\log p_{\phi'}(\hat{y}|\mathbf{x}_t)$ by the classifier parameters $\phi'$ that **do not** provides guidance throughout the sampling process.*

We set up the off-sampling classifier $\phi'$ with the same architecture and performance as the on-sampling classifier $\phi$ used for guidance. The only difference between the two models is the parameters. Off-sampling classifier is initialized as the parameters of the on-sampling classifier. We fine-tune the off-sampling model with 10000 timesteps with the same loss for training the on-sampling classifier. The testing accuracy between the off-sampling classifier and the on-sampling classifier is shown in Table 10 in Appendix D.

We visualize the *on-sampling* loss obtained from the noise-aware ADM classifier from Dhariwal & Nichol (2021) on ImageNet256x256, as shown in Figure 4. Our results indicate that classification information is predominantly active during the early stages and converges within the first 120 timesteps. In contrast, the *off-sampling* loss follows a different trend, converging only after the denoising process is nearly ended. This observation suggests that generated samples behave inconsistently when evaluated with classifiers of similar performance but different parameters, highlighting an over-adjustment to the model's parameters rather than to the true characteristics or conditions of the generated images.

**Definition 3.** *Model-fitting occurs when sampled images $\mathbf{x}_t$ at timestep $t$ is updated to maximize $p_\phi(y|\mathbf{x}_t)$ or to satisfy the parameters of the $\phi$ only instead of the real distribution $q(y|\mathbf{x}_t)$.*

In practice, a pretrained $p_\phi(y|\mathbf{x}_t)$ is only able to capture part of the $q(y|\mathbf{x}_t)$. Fitting solely with $p_\phi(y|\mathbf{x}_t)$ limits the sample's generalisation ability, leading to incorrect features or overemphasising certain details due to misclassification or overfocusing of the guidance classifier. Three pieces of evidence support that the vanilla guidance suffers from **model-fitting** problem.

**Evidence 1:** From Fig.4, we see that while the on-sampling loss converges around the $120^{th}$ timestep, the off-sampling loss remains high until the diffusion model converges later. This indicates that samples $\mathbf{x}_t$ at timestep $t$ satisfy only the on-sampling classifier but not the off-sampling classifier, despite their identical performance and architecture. Although the off-sampling loss decreases by the end, a significant gap between the off-sampling and on-sampling losses persists. This supports our hypothesis that the guidance sampling process produces features that fit only the guidance classifier, not the conditional information.

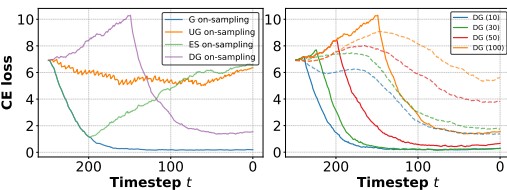

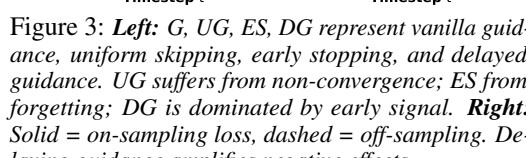

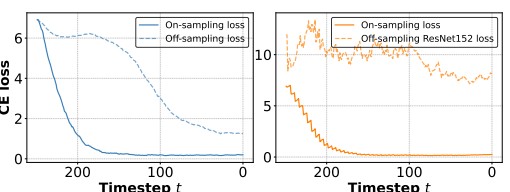

Figure 3: **Left:** *G, UG, ES, DG represent vanilla guidance, uniform skipping, early stopping, and delayed guidance. UG suffers from non-convergence; ES from forgetting; DG is dominated by early signal.* **Right:** *Solid = on-sampling loss, dashed = off-sampling. Delaying guidance amplifies negative effects.*

Figure 4: *(left) OADM-C, (right) Resnet152 off-sampling loss. The On-sampling loss converges very early while leaving the off-sampling loss converges at the end of the process after the conclusion of the denoising process.*

**Evidence 2:** Table 1 illustrates the model-fitting problem through accuracy metrics. With vanilla guidance, the accuracy is about 90.80% for the on-sampling classifier. However, the same samples evaluated by the off-sampling classifier or Resnet152 achieve only around 62.5% and 34.2% accuracy, respectively. This indicates that many features generated by the model are specific to the guidance classifier and do not generalize to other models.

Table 1: *A significant gap exists between the on-sampling and the off-sampling classifier in terms of accuracy, indicating model-fitting.*

| Evaluation Model | Accuracy |
|---|---|
| *On-sampling classifier* | 90.8% |
| *Off-sampling classifier* | 62.5% |
| *Off-sampling Resnet152* | 34.2% |

**Evidence 3:** Figure 2 (first row) shows samples from vanilla guidance, where every sampling step receives guidance information. Applying guidance at all timesteps forces the model to fit the on-sampling classifier's perception. Often, this makes the model colour-sensitive, focusing only on generating the "orange" feature for Goldfish and ignoring other details such as the shape, position and texture. From the three pieces of evidence we can observe, we can conclude that the vanilla guidance scheme has suffered from the model-fitting problem.

**Analogy to overfitting:** In neural network training, we have a dataset $\mathbf{x}$ and a classifier $f_\theta(\mathbf{x})$ to approximate the posterior distribution $p(y|\mathbf{x})$. Let $\mathbf{x}_{\text{train}}$ be the training data and $\mathbf{x}_{\text{test}}$ the testing data. Overfitting occurs when $f_\theta$ is tailored to fit $\mathbf{x}_{\text{train}}$ but fails to generalize to the entire dataset $\mathbf{x}$. This is observed by the gap between training loss/accuracy and testing loss/accuracy on $\mathbf{x}_{\text{train}}$ and $\mathbf{x}_{\text{test}}$.

Table 3: *Overfitting vs. Model-Fitting*

| Aspect | Overfitting | Model-fitting |
|---|---|---|
| **Train Data** | $\mathbf{x}_{\text{train}}$ | $f_{\phi_g}$ |
| **Test Data** | $\mathbf{x}_{\text{test}}$ | $f_{\phi_o}$ |
| **Parameters** | $f_\phi$ | $\mathbf{x}$ |

In the diffusion model's sampling process, the classifier $f_\phi$ is pretrained or fixed. The aim is to adjust the samples $\mathbf{x}$ to match the trained posterior $p_\phi(y|\mathbf{x})$. This process also uses Stochastic Gradient Descent with different roles: $f_\phi$ acts as the fixed data, and $\mathbf{x}$ are the trainable parameters. The model-fitting problem arises when $\mathbf{x}$ is adjusted to fit only the specific $f_\phi$ instead of generalizing conditional information. Here, $f_\phi$ is the on-sampling "data", and off-sampling "data" $f_{\phi'}$ is used to observe the model-fitting, analogous to using training and testing data for overfitting observation.

**Note: Similar model-fitting analysis for classifier-free guidance is in Appendix C.**

### 4.2 ANALYSIS

Gradient over-calculation is the main reason for model-fitting. Thus, **gradient balance**, which is not to call too many times of gradient calculation, is required. A straightforward solution is to eliminate the gradient calculations for the later timesteps, which have been found to be less active, as shown in Figure 4. This approach is referred to as Early Stopping (ES), where guidance is halted from the $200^{th}$ timestep onwards, continuing until the $0^{th}$ timestep.

**Early Stopping**: Figure 3 demonstrates that ES suffers from the *forgetting* problem, where on-sampling classification loss increases during the remaining sampling process, negatively impacting the generative outputs. This suggests that the guidance requires the property of **continuity**, meaning the gap between consecutive guidance steps must not be too large to prevent the *forgetting* problem.

**Uniform skipping guidance**: We tried an alternative approach named Uniform Skipping Guidance (UG). In UG, 50 guidance steps are evenly distributed across 250 sampling steps, with guidance

applied every five steps. This ensures continuity throughout the sampling process, mitigating the *forgetting* problem. However, as shown in Figure 4, UG encounters the issue of *non-convergence*, where the classification magnitude is too weak and becomes overshadowed by the denoising signals from the diffusion models, leading to poor conditional information. Thus, guidance must require another property, which is **magnitude sufficiency**.

**Delayed Guidance** (DG): Prior work Kynkäänniemi et al. (2025); Wang et al. (2024) suggests delaying guidance to avoid conflicts with the diffusion model. However, as shown in Figure 3 (right), longer delays worsen performance. *Why does this contradict Wang et al. (2024)?* That study assumes a conditional diffusion model, where applying guidance too early causes conflicts between guidance and conditional information of diffusion model. But in an unconditional model, there's no such conflict—so delaying guidance only harms performance.

In summary, vanilla guidance faces the issue of *model-fitting*, while ES and UG fail due to the *forgetting* and *non-convergence* problems, respectively. Therefore, the primary goal of our proposed method is to meet three key conditions which are **gradient balance**, **guidance continuity** and **magnitude sufficiency**.

## 4.3 Compress Guidance

To avoid calculating gradient too frequently, we propose to utilize the gradient from the previous guidance step at several next sampling steps, given that the gradient magnitude difference between two consecutive sampling steps is not too significant. By doing this, we can satisfy **magnitude sufficiency** without re-calculating the gradient at every sampling step. Note that the gradient directions have not been updated since the last guidance step, resulting in the **gradient balance**. Since all the sampling step receives a guidance signal, the **continuity** is guaranteed.

The hypothesis for utilizing the gradient from the previous timestep is three-fold. First of all, the avoidance of re-calculation of gradients frequently through the classifier prevents the generated samples from capturing the classification pattern of the classifier and helps to avoid model-fitting. Second, in the early stage, the avoidance of frequent gradient updates helps to avoid the noisy updated direction given by noisy samples. Finally, when the image is clear in the later timesteps, it is safe to skip the gradient calculation since the value of the gradient is less active during this stage as in Fig.4.

$$\mathbf{x}_{t-1} = \begin{cases} \mathbf{x}_t - \gamma_1 \nabla d - \gamma_2 \nabla D_{KL}[q(\hat{y}|\mathbf{x}_t)||q(y)], & \text{if } t \in G \\ \mathbf{x}_t - \gamma_1 \nabla d - \gamma_2 \Gamma_t, & \text{otherwise} \end{cases} \tag{11}$$

The set $G$ is the set of time-steps for which the gradient will be calculated. $\Gamma$ is a variable used to store the calculated gradient from the previous sampling step, $\Gamma_t$ is updated by $\Gamma_{t-1} = \begin{cases} \nabla D_{KL}[q(\hat{y}|\mathbf{x}_t) \| q(y)], & \text{if } t \in G \\ \Gamma_t, & \text{otherwise.} \end{cases}$. In practice, we find out that instead of duplicating gradients as in Eq. 11, we can slightly improve the performance by compressing the duplicated gradients into one guidance step instead of providing guidance to all sampling as in Eq.12. We name this method as *Compress Guidance*.We modify the sampling equation as below:

$$\mathbf{x}_{t-1} = \begin{cases} \mathbf{x}_t - \gamma_1 \nabla d - \gamma_2 \sum_{t=G_i}^{G_{i+1}} \Gamma_t, & \text{if } t = a_i \\ \mathbf{x}_t - \gamma_1 \nabla d, & \text{otherwise} \end{cases} \tag{12}$$

One of the algorithm's assumptions is that the magnitude is mostly the same for two consecutive sampling steps. From Appendix G, we observe that the classification gradient magnitude difference between two consecutive sampling steps is often larger in the early stage of the sampling process. Thus, we propose a method that distributes more guidance toward the early sampling stage and sparely at the end of the process. This will help to avoid the significant accumulation of magnitude differences in the early stage and help to deliver better performance as well as reduce the number of guidance steps. The scheme is defined as Eq. 13.

$$G_i = T - \lfloor \frac{T}{|G|^k} i^k \rfloor \quad \forall 0 \le i \le l, k \in [0; +\infty] \tag{13}$$

From the eq. 13, we have two main properties. First, when $k \to +\infty$, guidance timesteps are distributed toward the early stage of the sampling process. Second, when $k < 1$ and $k \to 0$, guidance

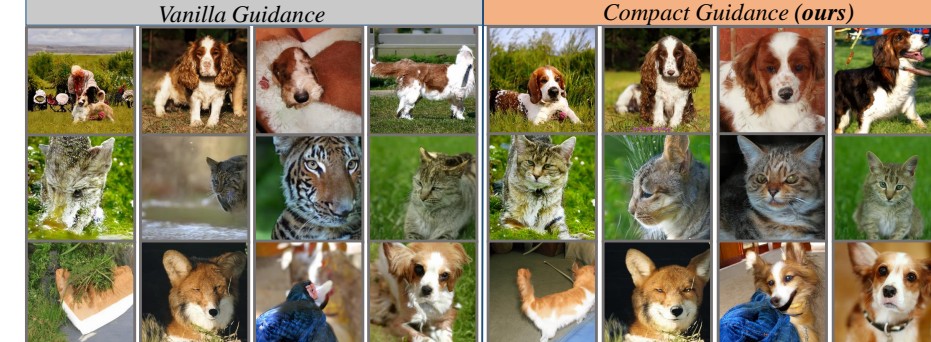

Figure 5: *ImageNet256x256. Left: Vanilla guidance applied at all timesteps. Right: Compress Guidance applied at 50 out of 250 timesteps. Compress Guidance reduces over-emphasized features, correcting weird and incorrect details. Further results are in AppendixH*

timesteps are distributed toward the late stage of the process. The proposed solution to select the timesteps for guidance as Eq.13 allows us to choose the number of timesteps we will do guidance and how to distribute these timesteps along the sampling process by adjusting the $k$ values. The full proof of these properties is written in the Appendix F. The full algorithm is in Algorithm 1 (Appendix).

### 4.4 COMPRESS GUIDANCE ON CLASSIFIER-FREE GUIDANCE

We start from the noise sampling equation of the classifier-free guidance as: $\tilde{\epsilon} = (1+w)\epsilon_\theta(\mathbf{x}_t, c, t) - w\epsilon_\theta(\mathbf{x}_t, t) = \epsilon_\theta(\mathbf{x}_t, c, t) + w(\epsilon_\theta(\mathbf{x}_t, c, t) - \epsilon_\theta(\mathbf{x}_t, t)) = \epsilon_\theta(\mathbf{x}_t, c, t) + wC$. $C$ could stand for classification information as mentioned in Dinh et al. (2023c). Replace the $\tilde{\epsilon}$ to Eq.9, we have:

$$\mathbf{x}_{t-1} = \mathbf{x}_t - \underbrace{(\frac{\sqrt{\alpha_t}-1}{\sqrt{\alpha_t}}\mathbf{x}_t + \frac{1-\alpha_t}{\sqrt{1-\bar{\alpha}_t}\sqrt{\alpha_t}}\epsilon_\theta(\mathbf{x}_t, c, t) - \sigma_t\mathbf{z})}_{\gamma_1\nabla d \quad \text{(match with Eq. 9)}} - \underbrace{\frac{\alpha_t-1}{\sqrt{1-\bar{\alpha}_t}}wC}_{\text{classification information}} \tag{14}$$

From this perspective, we can further apply the technique from Compress Guidance to the classification term in classifier-free guidance with the compression of classification information $\frac{\alpha_t-1}{\sqrt{1-\bar{\alpha}_t}}C$.

## 5 EXPERIMENTAL RESULTS

**Setup** Experiments are conducted on pretrained Diffusion models on *ImageNet 64x64*, *ImageNet 128x128*, *ImageNet 256x256*, *ImageNet 512x512* Deng et al. (2009) and *MSCOCO* Lin et al. (2014). The base Diffusion models utilized for label condition sampling task are ADM Dhariwal & Nichol (2021) and CADM Dhariwal & Nichol (2021) for classifier guidance, EDM2 Karras et al. (2023) DiTPeebles & Xie (2023) for classifier-free guidance (CFG) Ho & Salimans (2022), GLIDENichol et al. (2021) for CLIP text-to-image guidance and Stable Diffusion Rombach et al. (2022) for text-to-image classifier-free guidance. Other baselines we also do comparison is BigGAN Brock et al. (2018), VAQ-VAE-2 Zhao et al. (2020), LOGAN Wu et al. (2019), DCTransformers Nash et al. (2021). FID/sFID, Precision and Recall are utilized to evaluate image quality and diversity measurements. We denote Compress Guidance as "-CompG" and "-G" as vanilla guidance, "-CFG" is the CFG, and "-CompCFG" is our proposed Compress Guidance applying on CFG. Full results with details of the experimental set-up are discussed in Appendix D and E.

### 5.1 CLASSIFIER & CLASSIFIER-FREE GUIDANCE

Guidance in unconditional diffusion models enhances both image quality and diversity by providing conditional information during sampling, as shown in Table 4. CompressGuidance (CG) significantly improves FID, sFID, and Recall metrics, supported by qualitative evidence in Figures 5 and 11, and reduces guidance steps by 5×, leading to a 42% and 23% decrease in runtime on ImageNet 64x64 and 256x256, respectively (trade-off IS/FID can be observed in Fig. 8 and 9). In contrast, guidance in conditional diffusion models mainly boosts diversity, with smaller overall impact due to the model's inherent conditional structure. As shown in Table 13, CompG still improves Recall and reduces guidance steps by 5×, with notable runtime savings of 39.79%, 29.63%, and 22% on ImageNet

64x64, 128x128, and 256x256 resolutions, respectively. From section 4.4, we also apply the CompG technique on classifier-free guidance (CompCFG) and demonstrate the results in Table 5.

Table 4: *Unconditional guidance: CompG reduces guidance by 5× and improves performance.*

| Model | $\|G\|$ (↓) | GPU hours (↓) | FID (↓) | sFID (↓) | Prec (↑) | Rec (↑) |
|---|---|---|---|---|---|---|
| **ImageNet 64x64** | | | | | | |
| ADM (No guidance) | 0 | 26.33 | 9.95 | 6.58 | 0.60 | 0.65 |
| ADM-G | 250 | 54.86 | 6.40 | 9.67 | **0.73** | 0.54 |
| **ADM-CompG** | **50** | 31.80 | **5.91** | 8.26 | 0.71 | **0.56** |
| **ImageNet 256x256** | | | | | | |
| ADM (No guidance) | 0 | 245.37 | 26.21 | 6.35 | 0.61 | 0.63 |
| ADM-G | 250 | 334.25 | 11.96 | 10.28 | 0.75 | 0.45 |
| **ADM-CompG** | **50** | 258.33 | **11.65** | 8.52 | 0.75 | 0.48 |

Table 5: *Conditional diffusion: CompCFG yield lower FID and runtime with fewer guidance steps.*

| Model | $\|G\|$ (↓) | GPU hours (↓) | FID (↓) | sFID (↓) | Prec (↑) | Rec (↑) |
|---|---|---|---|---|---|---|
| **ImageNet 256x256** | | | | | | |
| DiT (No guidance) | 0 | 36.33 | 10.94 | 6.02 | 0.69 | 0.63 |
| DiT-CFG | 250 | 75.04 | 2.25 | **4.56** | 0.82 | 0.58 |
| **DiT-CompCFG** | **22** | 42.20 | **2.19** | 4.74 | **0.82** | 0.60 |
| **ImageNet 512x512** | | | | | | |
| EDM2 (No guidance) | 0 | 4.22 | 2.23 | 5.21 | 0.75 | 0.62 |
| EDM2-CFG | 32 | 8.63 | 1.84 | 4.06 | **0.83** | 0.59 |
| **EDM2-CompCFG** | **6** | 5.06 | **1.63** | 3.91 | 0.80 | **0.61** |

## 5.2 Text-to-Image Guidance

Table 6: *Stable Diffusion on **MSCOCO 256x256**. CompG improves quality (Fig. 1) and all metrics.*

| Model | $\|G\|$ (↓) | GPU hrs (↓) | FID (↓) | IS (↑) | CLIP (↑) | GenEval (↑) |
|---|---|---|---|---|---|---|
| SD-CFG | 50 | 54 | 16.04 | 32.34 | 30 | 0.42 |
| **SD-CompCFG** | **8** | **35** | **14.04** | **35.90** | **31** | **0.43** |

We apply the CompG on this task with two types of guidances, which are CLIP-based guidance (GLIDE) Nichol et al. (2021) and classifier-free guidance (Stable Diffusion) Rombach et al. (2022). The results are shown in Table 12 and 6 and Figure 1.

## 5.3 Ablation study

**Distribution guidance timesteps toward the early stage of the process:** According to the eq. 13, by adjusting $k$, we can distribute the timesteps toward the early stage or the late stage of the sampling process. Table 7 shows the comparison between $k$ values. With $k = 1.0$, guidance steps are distributed uniformly. Larger $k$ results in comparable performance but more fruitful running time and the number of guidance steps.

**Trade-off between computation and image quality** Compact rate is the total number of sampling steps over the number of guidance steps $\frac{T}{\|G\|}$. The larger the compact rate, the lower the model's guidance, hence the lower running time. Figure 12 shows the effect of fewer timesteps on IS, FID and Recall as in Figure 12a, 12b and 12c.

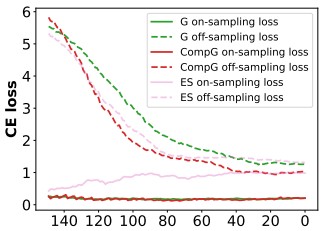

Figure 6: *From 150 to 250 sampling steps. CompG narrows the loss gap, mitigating overfitting. ES halts at 50 steps, leading to forgetting problem and loss increase.*

Table 7: **ImageNet64x64.** *Varying k shows improved efficiency and quality with fewer guidance steps and lower compute.*

| Model | $k$ | $\|G\|$ (↓) | GPU hours (↓) | FID (↓) | sFID (↓) | Prec (↑) | Rec (↑) |
|---|---|---|---|---|---|---|---|
| CADM (No guidance) | - | 0 | 26.64 | 2.07 | 4.29 | 0.73 | 0.63 |
| CADM-CompG | 1.0 | 50 | 32.22 | 1.91 | 4.38 | **0.77** | 0.61 |
| CADM-CompG | 5.0 | 32 | 29.81 | **1.82** | 4.31 | 0.76 | **0.62** |
| CADM-CompG | 6.0 | 28 | **29.12** | 1.93 | 4.35 | 0.75 | 0.62 |

Table 8: **ImageNet512x512.** *Interval guidance with CompCFG improves performance and diversity while reducing steps.*

| Model | $\|G\|$ (↓) | Guidance Interval | FID (↓) | sFID (↓) | Prec (↑) | Rec (↑) |
|---|---|---|---|---|---|---|
| EDM2-IntG | 6 | [17, 22] | 1.44 | 3.91 | 0.81 | 0.61 |
| EDM2-CompCFG | 6 | [17, +∞] | 1.44 | 3.88 | 0.81 | 0.62 |
| EDM2-CompCFG | **5** | [17, +∞] | 1.44 | **3.86** | 0.81 | **0.63** |
| EDM2-CompCFG | **4** | [17, +∞] | 1.45 | 3.87 | 0.80 | **0.63** |

**Comparison with other guidance variants**: Table 8 compares our proposed CompCFG with Interval Guidance methods from Kynkäänniemi et al. (2025); Wang et al. (2024). CompCFG achieves results comparable to IntG in Kynkäänniemi et al. (2025), but with broader applicability. Unlike IntG, which is limited to conditional diffusion models or classifier-free guidance, CompCFG can be integrated into any diffusion model, delivering improved image quality and computational cost, as demonstrated in Tables 4 and 5. Comparision with Dinh et al. (2023a;b); Zheng et al. (2022) is in Appendix E.1.

## 6 Conclusion

This paper quantifies model-fitting in diffusion model sampling, analogous to overfitting phenomenon, by analyzing on- and off-sampling loss. To address this, we propose Compress Guidance, which enhances generative performance while reducing guidance steps by at least fivefold and cutting runtime by approximately 40%. Broader impacts and safeguards are discussed in Appendix A.

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

---

**Algorithm 1** Compress Guidance

---

**Input:** class labels $y$, classification scale $s$
$\mathbf{x}_T \sim \mathcal{N}(\mathbf{0}, \mathbf{I})$
$\Gamma \leftarrow 0$
$G \leftarrow$ Using Eq.13
**for** $t = T, ..., 1$ **do**
    $z \sim \mathcal{N}(\mathbf{0}, \mathbf{I})$
    **if** $t \in G$ **then**
        $g \leftarrow s\nabla_{\mathbf{x}_t} \log p_\phi(y|\mathbf{x}_t)$
        $G'_t \leftarrow$ the next guidance step
        $\Gamma \leftarrow g \times |t - G'_t|$
        $\mathbf{x}_{t-1} \leftarrow \frac{1}{\sqrt{\alpha_t}}(\mathbf{x}_t - \frac{1-\alpha_t}{\sqrt{1-\bar{\alpha}_t}}\epsilon_\theta(\mathbf{x}_t, t)) + \sigma_t^2\Gamma + \sigma_t z$
    **else**
        $\mathbf{x}_{t-1} \leftarrow \frac{1}{\sqrt{\alpha_t}}(\mathbf{x}_t - \frac{1-\alpha_t}{\sqrt{1-\bar{\alpha}_t}}\epsilon_\theta(\mathbf{x}_t, t)) + \sigma_t z$
    **end if**
**end for**

---

## A  BROADER IMPACT AND SAFEGUARD

The work does not have concerns about safeguarding since it does not utilize the training data. The paper only utilizes the pre-trained models from DiT Peebles & Xie (2023), ADMDhariwal & Nichol (2021), GLIDE Nichol et al. (2021) and Stable Diffusion Rombach et al. (2022). The work fastens the sampling process of the diffusion model and contributes to the population of the diffusion model in reality. However, the negative impact might be on the research on a generative model where bad people use that to fake videos or images.

## B  FULL ALGORITHMS

Algorithm 1 shows full algorithm. The full source code will be released beyond acceptance.

## C  MODEL-FITTING ANALYSIS FOR CLASSIFIER-FREE GUIDANCE

Different from classifier guidance, classifier-free guidance does not have an explicit classifer loss. However, that does not mean that classifier-free guidance does not suffer from model-fitting. We expand the analysis to CFG using the following steps:

1. Define the classifier in CFG

2. Define the observable loss

3. Observe on/off sampling

**Define the classifier in CFG** Alexander *et al.* Li et al. (2023) proved that a conditional diffusion model is itself a classifier with the classification objective:

$$\arg\min_c \ \mathbb{E}_{t,\epsilon}\big[\|\epsilon_\theta(x_t, c) - \epsilon(x_t)\|\big], \tag{15}$$

where $x_t$ is a noisy version of $x_0$ with random noise $\epsilon(x_t)$.

Given this objective, we similarly define the objective function to "optimize" the parameter $x_t$, as

$$x_t^\star = \arg\min_{x_t} \ \Delta_c(x_t) \tag{16}$$

$$\text{s.t.} \quad \Delta_c(x_t) \leq \Delta_{c'}(x_t), \quad \forall c' \neq c, \ c' \in \mathcal{C}, \tag{17}$$

with

$$\Delta_{c'}(x_t) = \mathbb{E}_\epsilon\Big[\|\epsilon_\theta(x_t, c') - \epsilon(x_t)\|_2^2\Big], \quad c' \in \mathcal{C}, \tag{18}$$

as the logit for each class $c'$.

When denoising from $x_t$, we normally do not have $\epsilon(x_t)$. We replace this with the predicted $\epsilon_\theta(x_t)$, giving

$$\Delta_{c'}(x_t) = \mathbb{E}_\epsilon \left[ \|\epsilon_\theta(x_t, c') - \epsilon_\theta(x_t)\|_2^2 \right]. \tag{19}$$

Thus, the diffusion model itself acts as a classifier.

**Define observable loss:** We convert the objective into the observable cross-entropy loss as follows:

$$p(c' \mid x_t) = \frac{\exp(-\Delta_{c'}(x_t))}{\sum_{k \in \mathcal{C}} \exp(-\Delta_k(x_t))} \quad \implies \quad p(c \mid x_t) = \mathrm{softmax}(-\Delta(x_t))_c. \tag{20}$$

$$L_{\mathrm{CE}}(x_t, c) = -\log p(c \mid x_t) = \Delta_c(x_t) + \log \sum_{k \in \mathcal{C}} \exp\big(-\Delta_k(x_t)\big). \tag{21}$$

**On/Off sampling observation** We observe On/Off sampling loss using $L_{\mathrm{CE}}$. We use two public models from EDM2[1] which share the same architecture but differ in training hyperparameters: EDM-S-0.025 and EDM-S-0.085. Their performance is reported below:

| Model | FID | Accuracy |
|---|---|---|
| EDM-S-0.025 | 2.29 | 61% |
| EDM-S-0.085 | 2.40 | 63% |

Table 9: General performance of diffusion models on generative task (FID) and classification task (Accuracy).

The two models share similar performance. EDM-S-0.025 is used for on-sampling loss observation (joined in the guidance process), and EDM-S-0.085 is used for off-sampling observation (not used for guidance).

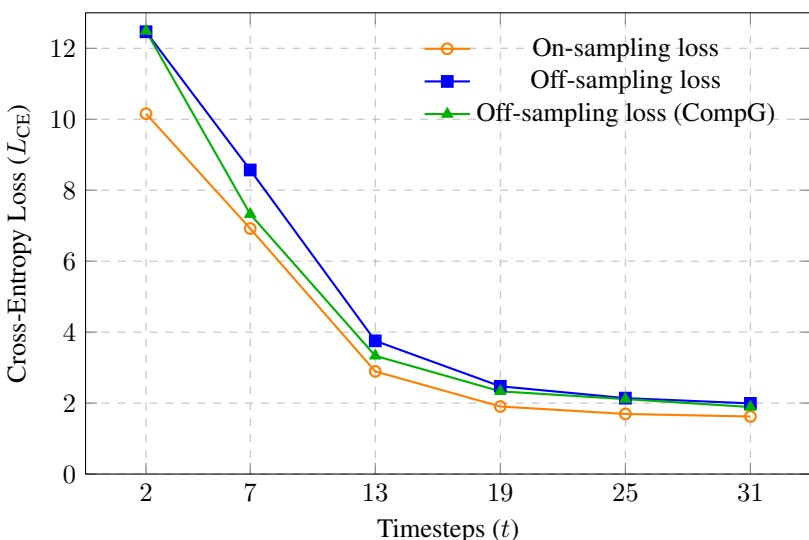

Figure 7: On-sampling and off-sampling loss across timesteps for EDM-S-0.025 and EDM-S-0.085 models. The On-sampling loss has a significant gap to the off-sampling loss. However, using our proposed CompressGuidance (CompG) helps to close the gap between On/Off sampling loss.

The results show that model-fitting also occurs for CFG. Given the same diffusion models with similar performance, the diffusion model used for guidance achieves much lower loss during sampling. The use of CompG narrows the gap between off-sampling and on-sampling loss, indicating reduced model-fitting. Furthermore, as shown in Section 5.3, our method significantly improves both runtime and sampling quality.

---

[1] https://shorturl.at/uJxeV

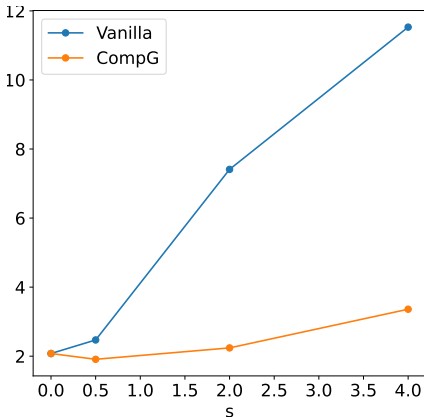

Figure 8: *FID curve given different guidance weight. This shows that the original vanilla guidance trades the quality with the diversity very significantly, while CompG helps to achieve the stability of the output.*

## D    EXPERIMENTAL SETUP

**Off-sampling classifier**: Off-sampling classifier is initialized as the parameters of the on-sampling classifier. We fine-tune the model with 10000 timesteps with the same loss for training the on-sampling classifier. The testing accuracy between the off-sampling classifier and the on-sampling classifier is shown in Table 10

| Evaluation Model | Accuracy |
|---|---|
| *On-sampling classifier* | 64.5% |
| *Off-sampling classifier* | 63.5% |

Table 10: Evaluation of On-sampling classifier and Off-sampling classifier on ground-truth images.

Figure 11 shows all the hyperparameters used for all experiments in the paper. Normally, since we skip a lot of timesteps that do guidance, the process will fall into the case of forgetting. To avoid this situation, we would increase the guidance scale significantly. The value of the guidance scale is often based on the compact rate $\frac{T}{|G|}$. A larger compact rate also indicates a larger guidance scale. In Table 15 and Figure 6, to achieve a fair comparison, we tune the guidance scale of CompG to achieve a similar Recall value with vanilla guidance. The reason is that the higher the level of diversity, the harder features can be recognized, resulting in higher loss and lower accuracy. If we don't configure similar diversity between the two schemes, the one with higher diversity will always achieve lower accuracy and higher loss value. We want to avoid the case that the model only samples one good image for all.

For all the tables, the models which are in bold are the proposed.

**GPU hours**: All the GPU hours are calculated based on the time for sampling 50000 samples in ImageNet or 30000 samples in MSCoco.

All experiments are run on a cluster with 4 V100 GPUs.

## E    FULL COMPARISON

Table 13 shows the full comparison with different famous baselines.

The stability of the CompG is visualized in Figure 8.

### E.1    ADDITIONAL ABLATION

In addition to Kynkäänniemi et al. (2025), one of the most recent studies on guidance, we compare our proposed method with Dinh et al. (2023a;b); Zheng et al. (2022). Dinh et al. (2023a) addresses

| MODEL | DATASET | $k$ | $s$ | $|G|$ | TIME-STEPS |
|---|---|---|---|---|---|
| **TABLE 4** | | | | | |
| ADM | IMAGENET 64x64 | - | 0.0 | 0 | 250 |
| ADM-G | IMAGENET 64x64 | - | 4.0 | 250 | 250 |
| ADM-COMPG | IMAGENET 64x64 | 1.0 | 4.0 | 50 | 250 |
| ADM | IMAGENET 256x256 | - | 0.0 | 0 | 250 |
| ADM-G | IMAGENET 256x256 | - | 4.0 | 250 | 250 |
| ADM-COMPG | IMAGENET 256x256 | 1.0 | 4.0 | 50 | 250 |
| **TABLE 5 & 13** | | | | | |
| CADM | IMAGENET 64x64 | - | 0.0 | 0 | 250 |
| CADM-G | IMAGENET 64x64 | - | 0.5 | 250 | 250 |
| CADM-COMPG | IMAGENET 64x64 | 1.0 | 2.0 | 50 | 250 |
| CADM-CFG | IMAGENET 64x64 | - | 0.1 | 250 | 250 |
| CADM-COMPCFG | IMAGENET 64x64 | 5.0 | 0.1 | 25 | 250 |
| CADM | IMAGENET 128x128 | 0.9 | 0.0 | 0 | 250 |
| CADM-G | IMAGENET 128x128 | - | 0.5 | 250 | 250 |
| CADM-CFG | IMAGENET 128x128 | - | 0.5 | 250 | 250 |
| CADM | IMAGENET 256x256 | - | 0.0 | 0 | 250 |
| CADM-G | IMAGENET 256x256 | - | 0.5 | 250 | 250 |
| CADM-COMPG | IMAGENET 256x256 | 1.5 | 0.5 | 50 | 250 |
| DIT-CFG | IMAGENET 256x256 | - | 1.5 | 250 | 250 |
| DIT-COMPCFG | IMAGENET 256x256 | 1.2 | 1.5 | 22 | 250 |
| EDM2-CFG | IMAGENET 256x256 | - | 1.2 | 32 | 32 |
| EDM2-COMPCFG | IMAGENET 512x512 | 2.5 | 0.3 | 6 | 32 |
| **TABLE 6 & 12** | | | | | |
| GLIDE-G | MSCOCO 64x64 | - | 0.0 | 250 | 250 |
| GLIDE-COMPG | MSCOCO 64x64 | 2.0 | 8.0 | 25 | 250 |
| GLIDE-G | MSCOCO 256x256 | - | 0.0 | 250 | 250 |
| GLIDE-COMPG | MSCOCO 256x256 | 2.0 | 5.5 | 35 | 250 |
| SDIFF-CFG | MSCOCO 256x256 | - | 2.0 (FID, IS),7.5 (CLIP, GENEVAL) | 50 | 50 |
| SDIFF-COMPCFG | MSCOCO 256x256 | 1.0 | 2.0(FID, IS), 7.5(CLIP, GENEVAL) | 8 | 50 |
| **TABLE 7** | | | | | |
| CADM | IMAGENET 64x64 | - | 0.0 | 0 | 250 |
| CADM-G | IMAGENET 64x64 | - | 4.0 | 250 | 250 |
| CADM-COMPG | IMAGENET 64x64 | 5.0, 6.0 | 4.0 | 50 | 250 |
| **TABLE 8** | | | | | |
| EDM2-INTG | IMAGENET 256x256 | - | 2.0 | 6 | 32 |
| EDM2-COMPCFG $|G| = 6$ | IMAGENET 512x512 | 2.5 | 1.6 | 6 | 32 |
| EDM2-COMPCFG $|G| = 5$ | IMAGENET 512x512 | 2.5 | 1.7 | 5 | 32 |
| EDM2-COMPCFG $|G| = 4$ | IMAGENET 512x512 | 2.5 | 1.7 | 4 | 32 |

Table 11: All hyper-parameters required for reproducing the results.

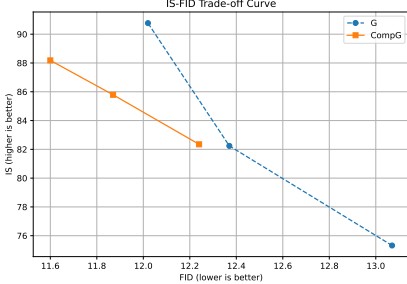

Figure 9: *IS-FID curve of ImageNet256x256. The performance shows that CompG comes with high and stable IS with improves in FID more significantly than vanilla G given IS increases..*

| Model | $|G|$ ($\downarrow$) | GPU hrs ($\downarrow$) | ZFID ($\downarrow$) |
|---|---|---|---|
| **MSCOCO 64x64** | | | |
| GLIDE-G | 250 | 34.04 | 24.78 |
| **GLIDE-CompG** | **25** | **20.93** | **24.5** |
| **MSCOCO 256x256** | | | |
| GLIDE-G | 250 | 66.84 | 34.78 |
| **GLIDE-CompG** | **35** | **37.55** | **33.12** |

Table 12: *Applying CompG on text-to-image GLIDE classifier-based guidance on MSCoco datasets.*

| Model | $|G|$ ($\downarrow$) | GPU hours ($\downarrow$) | FID ($\downarrow$) | sFID ($\downarrow$) | Prec ($\uparrow$) | Rec ($\uparrow$) |
|---|---|---|---|---|---|---|
| **ImageNet 64x64** | | | | | | |
| BigGAN | - | - | 4.06 | 3.96 | 0.79 | 0.48 |
| IDDPM | 0 | 28.32 | 2.90 | 3.78 | 0.73 | 0.62 |
| CADM (No guidance) | 0 | 26.64 | 2.07 | 4.29 | 0.73 | 0.63 |
| CADM-G | 250 | 53.52 | 2.47 | 4.88 | **0.80** | 0.57 |
| **CADM-CompG** | **50** | **32.22** | **1.91** | **4.57** | 0.77 | **0.61** |
| CADM-CFG | 250 | 54.97 | 1.89 | 4.45 | **0.77** | 0.60 |
| **CADM-CompCFG** | **25** | **29.29** | **1.84** | **4.38** | 0.77 | **0.61** |
| **ImageNet 128x128** | | | | | | |
| BigGAN | - | - | 6.02 | 7.18 | 0.86 | 0.35 |
| LOGAN | - | - | 3.36 | - | - | - |
| CADM (No guidance) | 0 | 61.60 | 6.14 | 4.96 | 0.69 | 0.65 |
| CADM-G | 250 | 94.06 | 2.95 | 5.45 | **0.81** | 0.54 |
| **CADM-CompG** | **50** | **66.19** | **2.86** | **5.29** | 0.79 | **0.58** |
| **ImageNet 256x256** | | | | | | |
| BigGAN | - | - | 7.03 | 7.29 | 0.87 | 0.27 |
| DCTrans | - | - | 36.51 | 8.24 | 0.36 | 0.67 |
| VQ-VAE-2 | - | - | 31.11 | 17.38 | 0.36 | 0.57 |
| IDDPM | - | - | 12.26 | 5.42 | 0.70 | 0.62 |
| CADM (No guidance) | 0 | 240.33 | 10.94 | 6.02 | 0.69 | 0.63 |
| CADM-G | 250 | 336.05 | 4.58 | **5.21** | 0.81 | 0.51 |
| **CADM-CompG** | **50** | **259.25** | **4.52** | 5.29 | **0.82** | **0.51** |
| DiT-CFG | 250 | 75.04 | 2.25 | **4.56** | 0.82 | 0.58 |
| **DiT-CompCFG** | **22** | **42.20** | **2.19** | 4.74 | **0.82** | **0.60** |

Table 13: *We show full results of the model compared to other models not related to guidance.*

the conflict between denoising signals and guidance signals, similar to Wang et al. (2024); Dinh et al. (2023b) identifies adversarial features and mitigates them by reducing uncertainty; and Zheng et al. (2022) tackles the gradient vanishing issue in classifier guidance by adapting the guidance weight. While all of these methods are training-free, our proposed CompG is the first to focus on reducing guidance-related computational costs by identifying and eliminating redundant guidance steps during sampling. Our results demonstrate that CompG achieves the best FID while maintaining the lowest running time. The comparative results are presented in Figure 14.

| Model | $|G|$ ($\downarrow$) | GPU hours ($\downarrow$) | FID ($\downarrow$) | sFID ($\downarrow$) | Prec ($\uparrow$) | Rec ($\uparrow$) |
|---|---|---|---|---|---|---|
| **ImageNet 64x64** | | | | | | |
| CADM (No guidance) | 0 | 26.64 | 2.07 | 4.29 | 0.73 | 0.63 |
| CADM-G | 250 | 53.52 | 2.47 | 4.88 | **0.80** | 0.57 |
| CADM-ProG Dinh et al. (2023b) | 250 | 53.60 | 1.87 | 4.33 | 0.77 | 0.60 |
| CADM-PxP Dinh et al. (2023a) | 250 | 54.32 | 1.84 | **3.97** | 0.76 | 0.60 |
| CADM-EDS Zheng et al. (2022) | 250 | 53.23 | 1.85 | 4.36 | 0.76 | **0.61** |
| **CADM-CompG** | **50** | **32.22** | **1.82** | 4.31 | 0.76 | **0.61** |
| **CADM-CompCFG** | **25** | **29.29** | **1.84** | 4.38 | 0.77 | **0.61** |

Table 14: *Comparing CompG and CompCFG with other variants Dinh et al. (2023a;b); Zheng et al. (2022) of classifier guidance on conditional diffusion model ADM Dhariwal & Nichol (2021)*

| Guidance | On-samp. | Off-samp. | Resnet | FID |
|---|---|---|---|---|
| Vanilla | 90.8 | 62.5 | 34.17 | 2.47 |
| Early Stopping | 63.05 | 55.22 | 33.55 | 2.21 |
| CompG (ours) | **91.2** | **64.2** | **34.93** | **1.82** |

Table 15: *Model-fitting on **ImageNet64x64** samples. ES suffers from the forgetting problem and has low performance. CompG achieves higher both on on-sampling and off-sampling acc.*

## F   MATHEMATICAL DETAILS

**Proof of Theorem 1**

*Proof.* Given real data $\mathbf{x}_0$, at timestep $t$ we have $\mathbf{x}_t = \sqrt{\bar{\alpha}_t}\mathbf{x}_0 + \sqrt{1-\bar{\alpha}_t}\epsilon$. On the other hand, the prediction of real data has the form $\tilde{\mathbf{x}}_0^{(t)} = \frac{\mathbf{x}_t - \sqrt{1-\bar{\alpha}_t}\epsilon_\theta(\mathbf{x}_t,t)}{\sqrt{\bar{\alpha}_t}}$, replace $\mathbf{x}_t$ with $\mathbf{x}_0$ and $\epsilon$ we have $\tilde{\mathbf{x}}_0^{(t)} = \mathbf{x}_0 + \frac{\sqrt{1-\bar{\alpha}_t}(\epsilon - \epsilon_\theta(\mathbf{x}_t,t))}{\sqrt{\bar{\alpha}_t}}$. Thus, $||\tilde{\mathbf{x}}_0^{(t)} - \mathbf{x}_0|| = \frac{1-\bar{\alpha}_t||\epsilon - \epsilon_\theta(\mathbf{x}_t,t)||}{\bar{\alpha}_t}$  $\square$

If we further assume that $q(\mathbf{x}_0)$ has a form of Normal Distribution, we would have the final objective as $D_{KL}(q(\mathbf{x}_0)||p_\theta(\tilde{\mathbf{x}}_0|\mathbf{x}_t))$. Since $q(\mathbf{x}_0)$ has the form of Gaussian, we can have the minimization of $||\tilde{\mathbf{x}}_0^{(t)} - \mathbf{x}_0||$ would result in the minimization of $||q(\tilde{\mathbf{x}}_0) - q(\mathbf{x}_0)|| = ||\frac{q(\tilde{\mathbf{x}}_0)q(\mathbf{x}_t|\tilde{\mathbf{x}}_0)}{q(\mathbf{x}_t)} - q(\mathbf{x}_0)||$ since $\tilde{\mathbf{x}}_0 \sim p_\theta(\tilde{\mathbf{x}}_0|\mathbf{x}_t)$ with a deterministic forward of $\mathbf{x}_t$ to $\epsilon_\theta$, we have $q(\tilde{\mathbf{x}}_0) \approx \frac{q(\tilde{\mathbf{x}}_0)q(\mathbf{x}_t|\tilde{\mathbf{x}}_0)}{q(\mathbf{x}_t)} = p_\theta(\tilde{\mathbf{x}}_0|\mathbf{x}_t)$. Assume we have two density functions: $p(\mathbf{x})$ and $q(\mathbf{x})$. The KL divergence between these two has the form:

$$\int_0^1 p(\mathbf{x}) \log \frac{p(\mathbf{x})}{q(\mathbf{x})} = \int_0^1 p(\mathbf{x}) \log(p(\mathbf{x})) - p(\mathbf{x}) \log(q(\mathbf{x})) d\mathbf{x} \tag{22}$$

$$= \int_0^1 p(\mathbf{x}) \log(p(\mathbf{x})) d\mathbf{x} - \tag{23}$$

$$\int_0^1 p(\mathbf{x}) \log(p(\mathbf{x})) + p(\mathbf{x}) \log((\frac{p(\mathbf{x})}{q(\mathbf{x})} - 1) + 1) d\mathbf{x}$$

$$= \int_0^1 -p(\mathbf{x}) \log((\frac{q(\mathbf{x})}{p(\mathbf{x})} - 1) + 1) d\mathbf{x} \tag{24}$$

$$= \int_0^1 -(q(\mathbf{x}) - p(\mathbf{x})) + (q(\mathbf{x}) - p(\mathbf{x}))^2 (\frac{1}{p(\mathbf{x})} - \frac{1}{q(\mathbf{x})}) d\mathbf{x} \tag{25}$$

$$\leq \int_0^1 (q(\mathbf{x}) - p(\mathbf{x}))^2 (\frac{1}{p(\mathbf{x})} - \frac{1}{q(\mathbf{x})}) d\mathbf{x} \tag{26}$$

$$\leq \int_0^1 (q(\mathbf{x}) - p(\mathbf{x}))^2 (\frac{1}{a} - \frac{1}{b}) d\mathbf{x} = \frac{b-a}{ab} ||p - q|| \tag{27}$$

Thus $D_{KL}(p(\mathbf{x})||q(\mathbf{x})) \leq \frac{b-a}{ab}||p - q||$

Base on this bound we would have the minimization of $||p_\theta(\tilde{\mathbf{x}}_0|\mathbf{x}_t) - q(\mathbf{x}_0)||$ is equivalent to the minimization of $D_{KL}(q(\mathbf{x}_0)||p_\theta(\tilde{\mathbf{x}}_0|\mathbf{x}_t))$.

**Proof of first property of eq. 13**

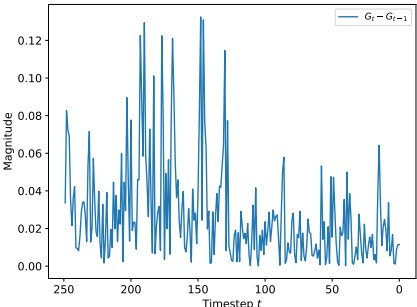

Figure 10: *Gradient magnitude difference measured at two consecutive steps*

*Proof.* Let $k_1 < k_2$ and $k_1, k_2 \in [1; +\infty]$, with $\frac{T}{|G|^k} i^k = T(\frac{i}{|G|})^k$ and $\frac{i}{|G|} < 1$, we have:

$$(\frac{i}{|G|})^{k_1} \geq (\frac{i}{|G|})^{k_2} \tag{28}$$

$$\Leftrightarrow T(\frac{i}{|G|})^{k_1} \geq T(\frac{i}{|G|})^{k_2} \tag{29}$$

$$\Leftrightarrow \lfloor T(\frac{i}{|G|})^{k_1} \rfloor \geq \lfloor T(\frac{i}{|G|})^{k_2} \rfloor \tag{30}$$

$$\Leftrightarrow T - \lfloor T(\frac{i}{|G|})^{k_1} \rfloor \leq T - \lfloor T(\frac{i}{|G|})^{k_2} \rfloor \tag{31}$$

As a result, $G_i^{(k_1)} \leq G_i^{(k_2)} \forall k_1, k_2 \geq 1$ and $k_1 < k_2$. With $k_2 \to +\infty$, $G_i^{(k_2)}$ is bounded by T. This means that larger $k$ values would result in the distribution of the timesteps toward the early stage of the sampling process. $\square$

**Proof of first property of eq. 13**

*Proof.* Similar to previous proof we have $G_i^{(k_1)} \leq G_i^{(k_2)} \forall k_1, k_2 \geq 1$ and $k_1 < k_2$. This also mean that $G_i^{(k_1)} > G_i^{(1)}$, $\forall 0 \leq k_1 < 1$ and if $k_1 \to 0$ then $G_i^{(k_1}$ $\to 0$, hence all the $g_i \in G^{(k_1)i}$ is bounded by 0. As a result, by adjusting $k$ toward 0, we would have the distribution of guidance steps toward the later stage of the sampling process $\square$

## G GRADIENT MAGNITUDE DIFFERENCE BETWEEN TWO CONSECUTIVE SAMPLING STEPS

In this section, we analyze the variation in the classification gradient throughout the sampling process, particularly its significant fluctuations during the early stages. To investigate this, we generate 32 images from the ImageNet64 dataset using ADM-G (Dhariwal & Nichol (2021)). The guidance classifier employed in this process is the noise-aware classifier trained within ADM-G. Our observations, illustrated in Figure 10, highlight how the classification gradient behaves over time, providing insights into its impact on the sampling process and model performance.

## H ADDITIONAL QUALITATIVE RESULTS

Due to space limitations in the main paper, we present qualitative results in this supplementary material. Figures 11, 13, 14, 15, and 16 provide additional comparisons with the vanilla baseline, while Figures 17 and 18 showcase high-quality images generated by DiT models combined with CompG.

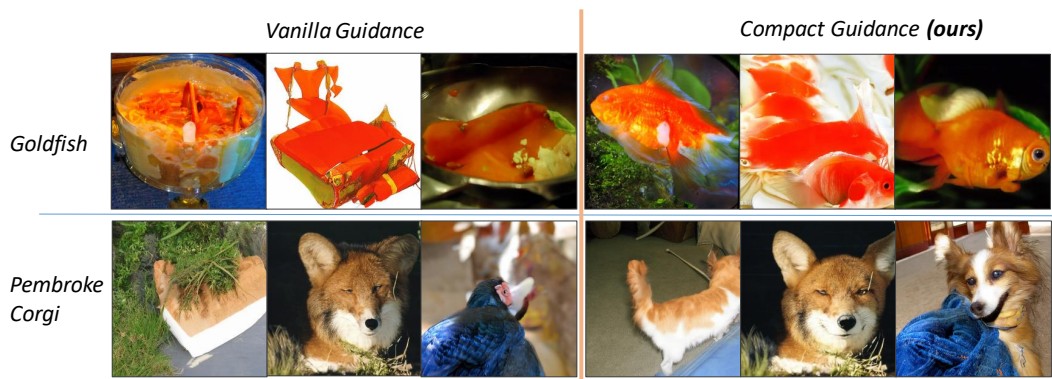

Figure 11: *Qualitative results on ImageNet256x256. Left: Vanilla guidance applied at all timesteps. Right: Compress Guidance applied at 50 of 250 timesteps. Compress Guidance corrects misclassification by the on-sampling classifier, preventing out-of-class image generation and restoring accurate class information. More qualitative results are shown in AppendixH*

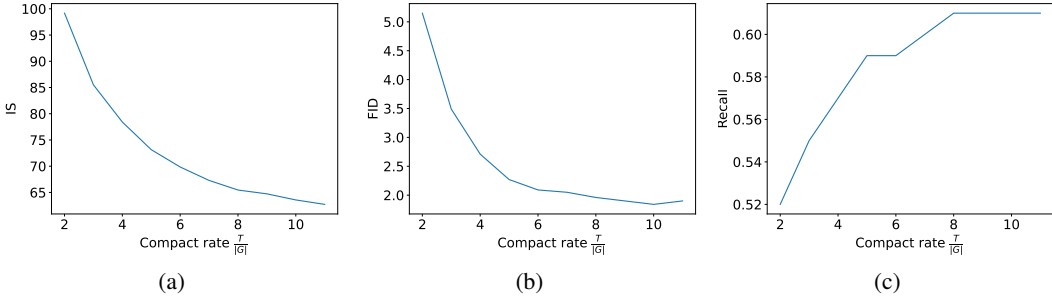

|     |     |     |
| :-: | :-: | :-: |
| (a) | (b) | (c) |

Figure 12: Trade-off: Running time versus performance. We measure the compact rate as $\frac{T}{|G|}$. In (a), IS decreases with increasing compact rate, while FID and Recall improve. However, when the rate exceeds 10, FID begins to rise. This suggests that increased diversity from more features initially enhances Recall and FID, but excessive diversity degrades image quality.

Quiet forest path surrounded by tall trees.

Beach at sunset with waves gently crashing.

StableDiffusion        (ours)

Figure 13: *Stable Diffusion with classifier-free guidance. The left figure is the vanilla classifier-free guidance with application on all 50 timesteps. Our proposed Compress Guidance method is the right figure, where we only apply guidance on 10 over 50 steps. The output shows our methods' superiority over classifier-free guidance regarding image quality, quantitative performance and efficiency.*

Serene mountain landscape with a clear sky

A white plate with breakfast foods on it

StableDiffusion    (ours)

Figure 14: *Stable Diffusion with classifier-free guidance. The left figure is the vanilla classifier-free guidance with application on all 50 timesteps. Our proposed Compress Guidance method is the right figure, where we only apply guidance on 10 over 50 steps. The output shows our methods' superiority over classifier-free guidance regarding image quality, quantitative performance and efficiency.*

Flowers are arranged in a vase sitting on a table.

A plate with food on it, a fork and some kind of drink

StableDiffusion      **(ours)**

Figure 15: *Stable Diffusion with classifier-free guidance. The left figure is the vanilla classifier-free guidance with application on all 50 timesteps. Our proposed Compress Guidance method is the right figure, where we only apply guidance on 10 over 50 steps. The output shows our methods' superiority over classifier-free guidance regarding image quality, quantitative performance and efficiency.*

1242
1243
1244
1245
1246
1247
1248
1249
1250
1251
1252
1253
1254
1255
1256
1257
1258
1259
1260
1261
1262
1263
1264
1265
1266
1267
1268
1269
1270
1271
1272
1273
1274
1275
1276
1277
1278
1279

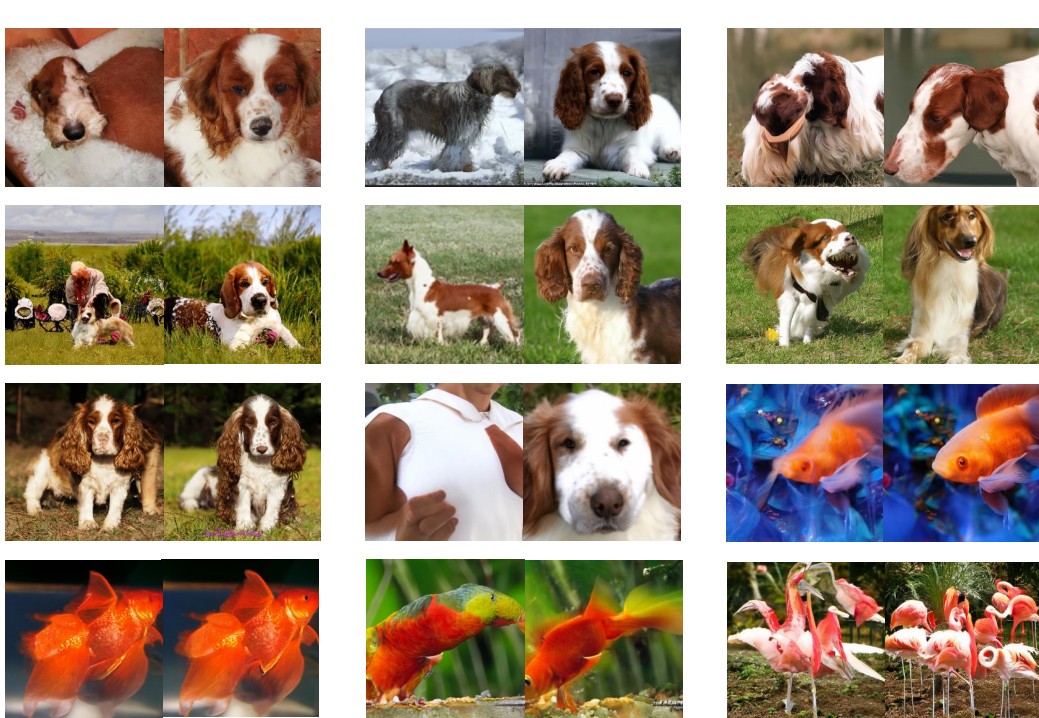

Figure 16: *Qualitative comparison between ADM-G and ADM-CompG.The images generated by ADM-G and ADM-CompG are put side by side. On the left side is ADM-G and on the right side is ADM-CompG.*

1280
1281
1282
1283
1284
1285
1286
1287
1288
1289
1290
1291
1292
1293
1294
1295

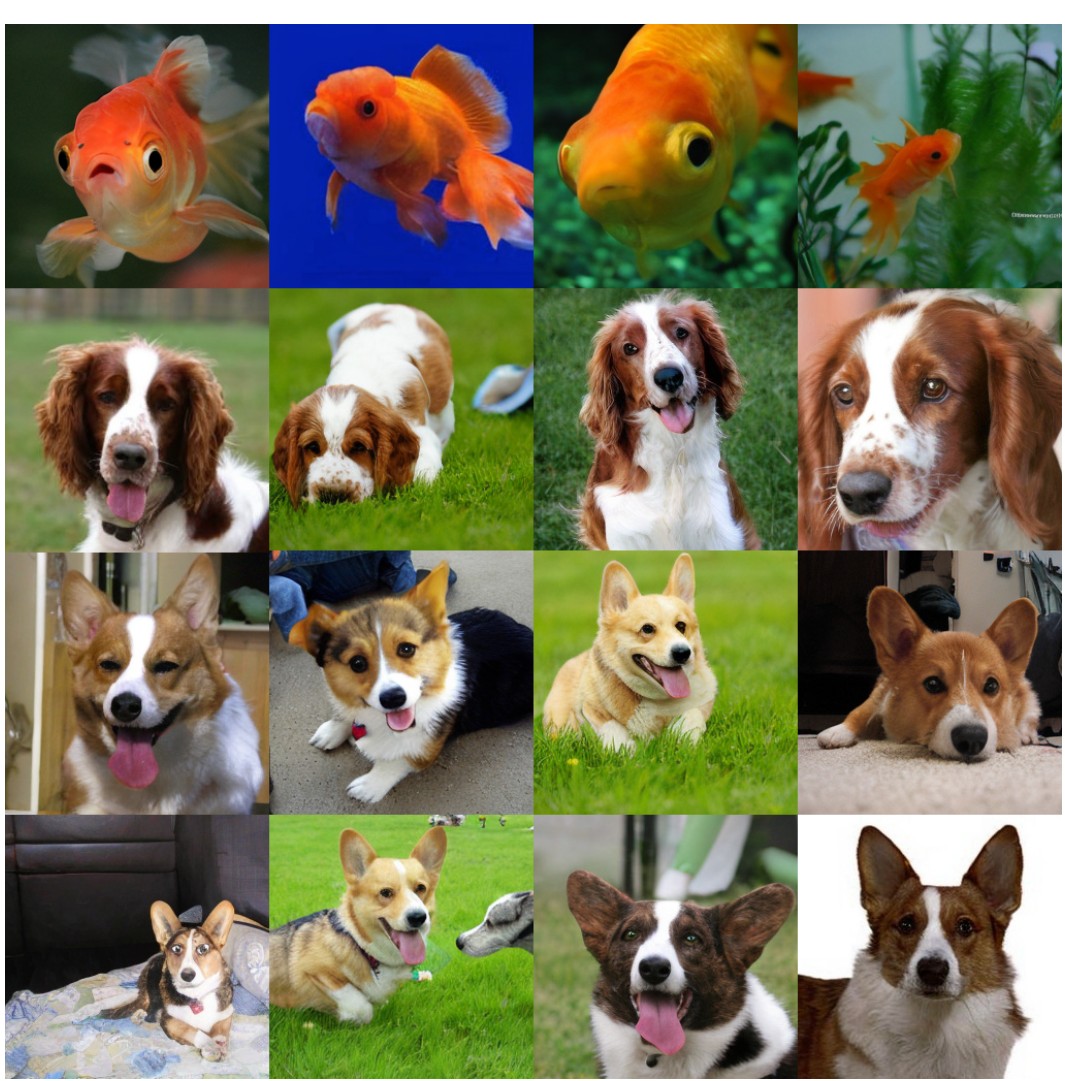

Figure 17: *Images generated by DiT-CompCFG. From top to bottom classes goldfish, Welsh springer spaniel, Pembroke Welsh corgi, Cardigan Welsh corgi.*

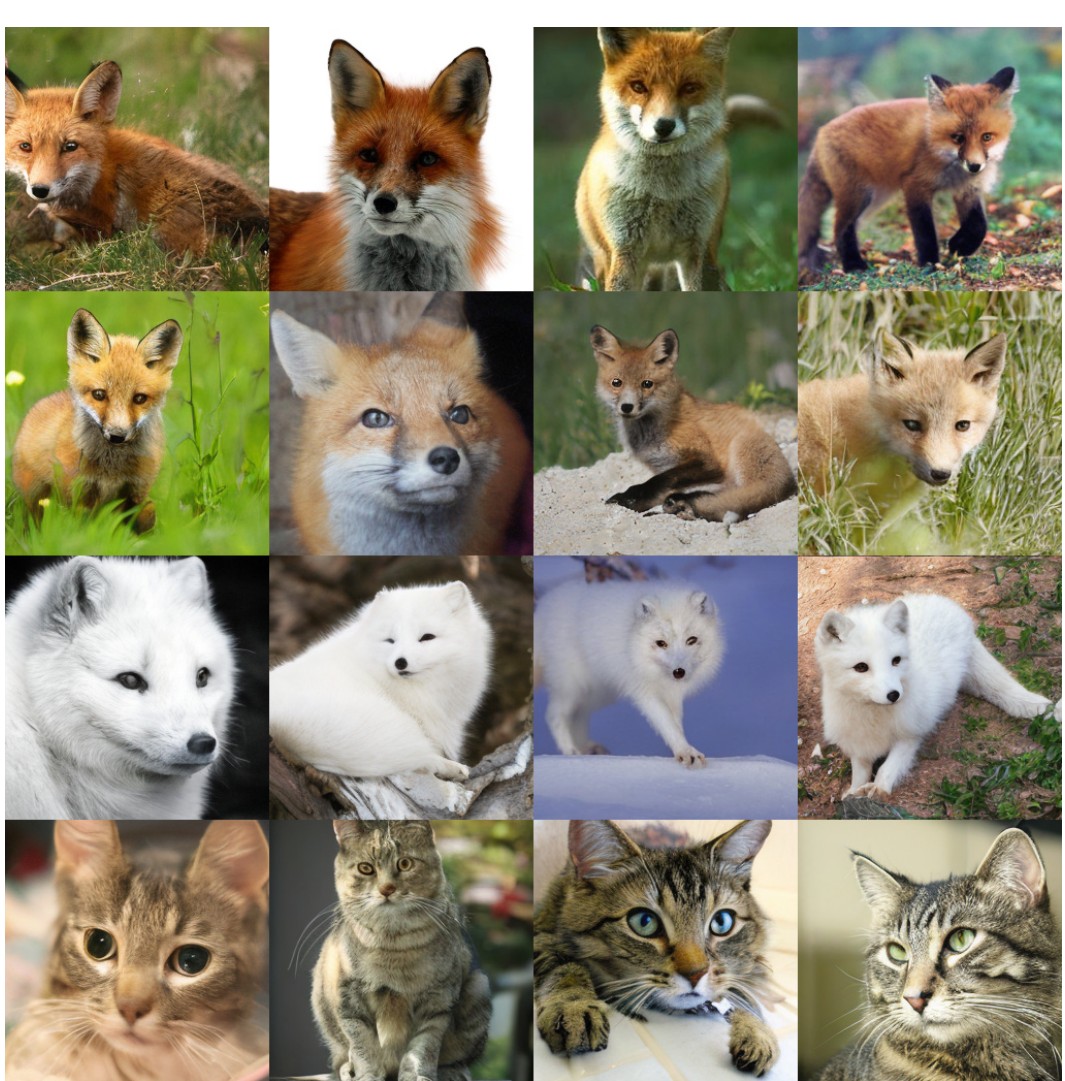

Figure 18: *Images generated by DiT-CompCFG. From top to bottom classes redfox, kitfox, Arctic fox, tabby cat.*