# OpenReview forum: "Fixing Model-Fitting: Compressing Guidance for Better Sampling"
_ICLR.cc/2026/Conference — Submitted to ICLR 2026_

### Official Review · Reviewer_TfJb · 2025-10-22

**Soundness:** 2
**Presentation:** 1
**Contribution:** 2
**Rating:** 2
**Confidence:** 4

**Summary:**

The paper analyses the phenomenon of overfitting in classifier-based guidance of diffusion models (including implicit classifiers in classifier-free guidance). One of the main findings is that guiding the generation with a classifier reduces the classifier's loss during the generation; however, this loss reduction does not generalize to comparable classifiers with different parameters. It indicates an overfitting-like behavior, which the paper calls model fitting. As a solution, Compress Guidance is proposed, which reduces the number of denoising steps with guidance and distributes more guiding steps to the early denoising steps. Evaluation of the method is conducted on traditional ImageNet diffusion models (ADM and EDM2), together with more advanced models like Stable Diffusion.

**Strengths:**

-	The paper tackles an interesting issue: classifier-based/CFG-based generations incorporate additional computational costs. By reducing the number of guidance steps, the computational burden can be reduced. Moreover, as the paper demonstrates, this strategy also improves image quality and prevents overfitting of the guidance model.
-	Qualitative and quantitative experiments support the motivation of the paper well, and sufficient mathematical foundations are provided.
-	Quantitative evaluations indicate improved image quality compared to vanilla guidance strategies.

**Weaknesses:**

-	The paper writing should be improved, making it sometimes hard to follow. Here are some examples:
  - The abstract directly starts with the term of model-fitting, never mentioning the setting of image generation with diffusion models.
  - L173: “With ˜x0 is the prediction of x0 at timestep t, we can…”
  - References to equations are sometimes abbreviated, e.g., Eq. 8 (L189), and sometimes not, e.g., equation 8 (L177)
  - L195: “where the xt is the parameter of the model at the timestep t,”-> why is x the parameter of the diffusion model here?
  - Figure 4 is referenced long before Figure 3. Also, there seems to be a wrong reference in L325 (should be Fig. 3 instead of 4, I assume)
  - L342: “To avoid calculating too much gradient”
  - Font sizes in tables are comparably small, e.g., Table 5. The same goes for font sizes in figures, such as Figs. 3 and 4.
-	Most experiments focus on highly outdated and simple ImageNet models (ADM from 2021). Whereas there are some qualitative experiments on Stable Diffusion in the Appendix and some numerical results in Table 6, the focus on outdated models limits the method’s strength. Classifier-guided generations are rarely used today, and CFG stands out as the go-to method for diffusion models.
-	Comparison to related guidance techniques is limited to Table 8. What about other settings, e.g., with Stable Diffusion/text-to-image domains?

Small remarks:

-	Metrics like FID should be cited.

**Questions:**

- Would it make sense to combine the method with some momentum or gradient decay, i.e., reducing the influence of gradients from the previous generation steps?

---

> ### Author Response · Authors · 2025-11-26
> **We have solved your concerns**
>
> We thank the reviewer for the thoughtful comments on our work. We have solved all of the concerns as below:
>
> ## W1. Writing problem
>
> We will fix those writing issues in the camera-ready version
>
> ## W2. Outdated ADM baseline:
> **Please note that beside ADM, we include EDM2 and DiT which are more modern architectures for diffusion**
>
> ### 1. Our baselines are widely selected for SOTA/popularity on different aspects.
>
> We first want to emphasise that our work focuses on improving guidance for generative models. When designing our experiments, our goal is to demonstrate that our method can be applied to a wide range of diffusion models. As a result, the logic behind our experimental design is as follows:
>
> **Diffusion-wise baselines:**
>
> > 1. Diffusion in pixel space (SOTA: ADM)
> > 2. Diffusion in latent space (DiT, EDM2)
>
> **Task-wise baselines:**
>
> > 1. Class generation (ADM, DiT, EDM2)
> > 2. Text-to-image generation (GLIDE, SD)
>
> **Guidance-wise baselines:**
>
> > 1. Classifier guidance (ADM, GLIDE)
> > 2. Classifier-free guidance (DiT, EDM2, SD)
>
> Therefore, although ADM is an older baseline, we still include it because it remains the SOTA model for pixel-space diffusion. We understand that the community has recently focused more on latent-based diffusion models. However, from a scientific perspective, we cannot simply follow trends. Our task is to propose a method that addresses model-fitting issues and show that the method is generalizable. Who knows—pixel-based generative models may become popular again in the future.
>
> ### 2. Why classifier guidance?
>
> Similar to the reasoning behind including ADM, we include classifier guidance in our experiments for four reasons:
>
>
> **1. Classifier guidance is beneficial for analysis.**
> Classifier-free guidance does not provide a classification loss, making it difficult to analyse model fitting. There are very limited ways to observe on-sampling loss and off-sampling loss under classifier-free guidance. In the paper, we use classifier guidance as a simple baseline for analysis, and we extend this analysis to classifier-free guidance in Section C of the Appendix. The observation of on/off-sampling loss for classifier-free guidance follows [R1], but the loss calculation is extremely expensive and hard to scale.
>
> **2. Classifier guidance is one of the two major types of guidance.**
> It is still important to shed light on classifier guidance when exploring the fundamental problems of guidance in diffusion models.
>
> **3. Classifier-free guidance is only useful in certain scenarios.**
> Classifier-free guidance assumes full labels for the entire dataset, which is unrealistic in many situations. In practice, we often only have labels for a small part of the data, making it impossible to obtain both conditional models required for classifier-free guidance. A much cheaper and more practical approach is to train a classifier on the labelled subset and use that classifier for guidance.
>
> **4. Classifier guidance could be extended in different forms that classifier-free guidance could not**
> In [R1], the authors extend classifier guidance into representative guidance, which can impossible to be done by classifier-free guidance.
>
>
>
> [R1] https://arxiv.org/abs/2303.16203
>
> [R2] https://openreview.net/pdf?id=gWgaypDBs8
>
>
> ## W3. Compare with other settings
>
> We have compared the method with other recent works on other CFG schedules, including Interval Guidance in Table 8 in the main paper (IntG is the current SOTA method for CFG). In [R3], the author also mentioned that other schedulers follow the same setup with a different scheduler as below:
>
> linear: $w(t) = 1 - t/T$
>
> sine: $w(t) = cos(\pi t/T) + 1$
>
> V-shape: $w(t) = invlinear(t)$, if $t < T/2$, $linear(t)$, else.
>
>
> The results are shown in Table R1
>
> | Model               | G  | GPU Hours ↓ | FID ↓ | sFID ↓ | Prec ↑ | Rec ↑ |
> |--------|----|-----|----|-----|----|---|
> | EDM2 (No guidance)  | 0  | 4.22        | 2.23  | 5.21   | 0.75   | 0.62  |
> | EDM2-CFG            | 32 | 8.63        | 1.84  | 4.06   | 0.83   | 0.59  |
> | EDM2-CFG (linear)   | 32 | 8.63        | 3.05  | 5.86   | 0.63   | 0.52  |
> | EDM2-CFG (sin)      | 32 | 8.63        | 2.58  | 5.88   | 0.68   | 0.50  |
> | EDM2-CFG (V-shape)  | 32 | 8.63        | 1.82  | 4.06   | 0.83   | 0.59  |
> | EDM2-CompCFG        | 6  | **5.06**        | **1.63**  | **3.91**   | 0.80   | **0.61**  |
> *Table. R6, the proposed CompCFG outperforms other schedulers significantly with much cheaper cost*
>
> [R3] Analysis of Classifier-Free Guidance Weight Schedulers
>
> ## W4. A combination of CompG and other methods, such as momentum and gradient decay
>
> Yes, we believe that the combination would be helpful in tackling issues in guidance, especially guidance based on a classifier. However, the combination of CompG and other methods, such as momentum, would require a rigorous design and analysis. We would take this comment for future work development.

---

> > ### Comment · Reviewer_TfJb · 2025-11-27
> >
> > I thank the authors for their response and the additional details.
> >
> > Regarding the baselines: I would not consider GLIDE or Stable Diffusion to be state of the art, particularly since it is unclear which version of Stable Diffusion was used. Nevertheless, I appreciate the clarification on model selection and find it acceptable.
> >
> > I also appreciate the additional details and results provided. I have decided to increase my score to 4. However, I want to emphasize that simply stating “We will fix those writing issues” is not sufficient. With the increasing number of conference submissions and the corresponding reviewing load, papers should meet a certain quality standard at the time of submission. While some minor mistakes are understandable and easily corrected, I believe the paper currently lacks sufficient writing quality and omits essential details, as outlined in my original review. Therefore, I am not willing to support acceptance of the paper. That said, I am happy to discuss this further with the other reviewers and the AC.

---

> ### Author Response · Authors · 2025-11-27
> **We thank the reviewer for recognizing our work.**
>
> Dear Reviewer Tfjb,
>
> Thank you a lot for your recognition of our work and increase your score. We know reviewing paper is a rigorous task, and your time spending on our paper is extremely valuable to us. We hope you will find our explanation acceptable and support our work.
>
> ## GLIDE/SD
> GLIDE is definitely not the SOTA for text2image generation. However, it is the only model doing the **text2image generation with classifier guidance**. For Stable Diffusion, we utilize Stable Diffusion 2.0. It might not be the SOTA, yet for this one we have a justification. The Text2Image task is a quickly moving pace, new model is released after several months coming with large number of parameters. A new released model also claims itself as SOTA. We, as researchers in academia, are not always be able to keep up with that pace.  As a result, we pick a typical model for evaluation, in this case we chose Stable Diffusion2.0.
>
> ## Writing
> Regarding the writing, we want to give some further elaboration:
> 1. The abstract directly starts with the term of model-fitting, never mentioning the setting of image generation with diffusion models.
> >  * From line 017-020: We did mention "we experiment on label-condition and text-to-image generation". We will re-write the phrase " label-condition image generation and text-to-image generation" to make it clearer.
> > * We understand that diffusion is applicable to video, audio and even text generation recently. *Then why we don't mention image generation task at first?*. The reason is that we are viewing the research on the view of fundamental question rather than application approach. Our work aims to solve the **model-fitting problem of diffusion** in general, and we benchmark on image generation. From this perspective, model-fitting and diffusion are the main objects need mentioning first.
> > * Different works that having the same approach with us also having the same writing that don't mention the generation task before the generation model such as Classifier-free guidance (Nips2021W, https://arxiv.org/pdf/2207.12598), Progressive Distillation (ICLR2022, https://arxiv.org/pdf/2202.00512), DMD2 (Nips 2024, https://arxiv.org/pdf/2405.14867).
> >> We will re-write the abstract to suit with both of the research views
> 2. L173: “With ˜x0 is the prediction of x0 at timestep t, we can…”
> > We will replace "With" by "Denoting". The reason why we use "with" is because we have already mentioned $x_0$ is an exact prediction of real data, as a result, we assume that $ \tilde{x} _0$ is the approximation prediction.
> >> We will fix this assumption in the paper.
> 3. References to equations are sometimes abbreviated, e.g., Eq. 8 (L189), and sometimes not, e.g., equation 8 (L177)
>  > (See next author's comments)
> 4. L195: “where the xt is the parameter of the model at the timestep t,”-> why is x the parameter of the diffusion model here?
> > x is the parameters (correct). explanation in the next author's comments.
> 5. Figure 4 is referenced long before Figure 3. Also, there seems to be a wrong reference in L325 (should be Fig. 3 instead of 4, I assume)
> > We will place the Figure 3 before Figure 4 and fix the wrong inference. The reason why Figure 3 and Figure 4 is misplaced is because of Latex
> 6. L342: “To avoid calculating too much gradient”
> > We will change to "to avoid over-calculating gradient"
> 7. Font sizes in tables are comparably small, e.g., Table 5. The same goes for font sizes in figures, such as Figs. 3 and 4.
> > The reason why we have small font size is because we need to compress all the contents into 9 pages. We already cut half of the experiments to put in Appendix already, but the content is still very large. We could solve this problem in camera ready version because it will have one more page.
>
> ## Rejection considering the large number of submissions
> "With the increasing number of conference submissions and the corresponding reviewing load, papers should meet a certain quality standard at the time of submission. While some minor mistakes are understandable and easily corrected, I believe the paper currently lacks sufficient writing quality and omits essential details, as outlined in my original review."
> > * There are three reasons as I have answered previously
> >> 1. Do not represent perspectives of different readers
> >> 2. Content compression into 9 pages (could be solved if camera ready version gives us one more page, or put parts of content to Appendix)
>
> > **We hope that paper submission is not just a competition. No matter how many submissions we have, the main criteria of a research paper should be in its contributions and novelty. We are also researchers, we could not always be able to write things that could touch to every readers, because when we do research our perspective is bias to our views. That is one of the reasons why the community needs peer reviewing. We hope the reviewer will consider supporting us.**

---

> ### Comment · Reviewer_TfJb · 2025-11-27
>
> Thank you for the update. I want to clarify one point: **my (borderline) recommendation to reject the paper is not due to the large number of submissions.** I review each submission individually and carefully. My concern is that papers should meet a certain baseline of quality at the time of submission. Submitting an insufficiently written or poorly formatted paper that lacks crucial details, and then addressing these issues only at the end of the review process, should not become the norm.
>
> While small errors are understandable, as outlined in my review, I found many parts of the paper difficult to follow or missing important details, as well as containing typos. The list I provided is only a small sample and not exhaustive. I believe this is a valid point of criticism. That said, it is not the main reason for my (slight) rejection lean; I have already raised the score because many writing-related issues have been improved.
>
> Regarding model evaluation, I understand that assessing the very latest models is not always feasible, and I did not expect evaluations of models like Flux.2. However, Stable Diffusion 2.0 dates back to November 2022. Three years is a long period in this field, with substantial architectural and methodological advances. At minimum, I would have expected evaluation on models from early this year.
>
> But I am happy to discuss my concerns with other reviewers after the rebuttal and are open to change my mind if the majority has another view on these issues.

---

> ### Author Response · Authors · 2025-11-28
> **We have just re-read again the comments**
>
> Dear Reviewer Tfjb,
>
> ## Writing
> While editing the rebuttal version, we re-read the comments to fix and find out some critical points are not about the bad writing. Here are two examples:
>
> 1. e.g., Eq. 8 (L189), and sometimes not, e.g., equation 8 (L177)
> > In Line 177, we say "In this equation 8..." meaning that we are referring directly to the equation right before it. While in L189, The Eq.8 is the abbrivation for equation 8. This is the correct writing. For line 177, We have three ways to write it which are "This equation 8", "The Eq.8" and "This equation". Due to the text is the explanation for the Eq.8 right above it, so we prefer using "This equation 8". I don't think this would be wrong to write like this.
>
> 2. L195: “where the xt is the parameter of the model at the timestep t,”-> why is x the parameter of the diffusion model here?
> > $x_t$ is the parameters of the model are already correct. The Eq.9 could be re-written as: $ x_{t-1} = x_t - \gamma_1 \Delta _{x_t} ||x_0 - \tilde{x} _0 || $ . This equation is correct.
> >>> This is the optimization equation to update $x_t$ itself to become $x_{t-1}$ given that diffusion model is fixed. Thus, $x_t$ is becoming the parameters. And this is one of the most important details in the subsection.
> >>> We believe that the reviewer misunderstand this point.
>
> As a results, we can see that there is currently no typos or serious grammar-mistake that have been pointed out. We sum up three points why reviewers find our work is hard to follow:
> 1. The writing of our work is a little bit bias toward fundamental view instead of application approach. (We are writing to balance both fundamental view and application view)
> 2. The reviewer overlooked the meaning of equation 9. This is an important detail in our argument. (We urge reviewers to re-read again this part).
> 3. The compression to 9 pages causes the density in the information, small font in captions of Figures. (We hope to have an additional page for camera ready or will put inference images into Appendix).
>
> **Note: If the reviewer finds any other unreasonable writing, we believe that it is not about our work is written sloppily. Instead, it comes from the different approaches between us and the reviewer to the problem. We hope the reviewer can point out all of the points that make the reviewer hard to follow our work, so we could improve our work's quality.**
>
> ## About Stable Diffusion 2.0
>
> When we come across a field like Text2Image generation, the most updated SOTA is one key that we and the reviewer both agree that it is infeasible to catch up. Another point should be its practicality also. According to our experience in both Academia and Industry, Stable Diffusion2.0 (2 years old) and Stable Diffusion XL (2 years old) is still very popular. Here are few reasons:
> 1. The running time/ memory consumption of these two models are reasonable. If memory is limited then SD2.0 will be utilized, otherwise, SDXL is utilized. SDXL is still popular backbone in many recent application research published this year (2025) for example LBM[R1], ICLight[R2],
> 2. Both SD2.0 and SDXL are not under non-commercial license make them still popular among industrial institutes. While newer models such as Flux-dev1.0 are under non-commercial license.
>
> [R1] https://github.com/gojasper/LBM  (ICCV2025)
>
> [R2] https://github.com/lllyasviel/IC-Light (ICLR2025)
>
> As a result, investigation on SD2.0 is not that outdated and still have impact for both academia and industry (Our GPU RAM is not enough to support SDXL sampling).
>
> From the two points above, we hope the reviewer will have a different look on our work. Right now, we still keep the paper version same as submission, after the reviewer can validate what we write here is correct, we will make a final edit.

---

> > ### Comment · Reviewer_TfJb · 2025-11-28
> >
> > Thank you for the additional details. I will keep them in mind during the post-rebuttal discussion.
> >
> > However, just to clarify, SDv2.0 is from November 24, 2022 (almost exactly 3 years from today). And in terms of running time, models like Flux-fast/flux-schnell (apache-2.0 licence) and other recent models are more efficient since they only require a few inference steps, whereas SD usually takes 50-100. So stating that SDv2.0 is still sota is simply wrong, and there are many more recent alternatives that beat the model in terms of runtime and image quality.

---

> ### Author Response · Authors · 2025-11-28
> **We thanks the reviewer for the response**
>
> Dear Reviewer Tfjb,
>
> First, we thank the reviewer for acknowledging the additional details. We hope that we have solved all the concerns related to the writing from the reviewer. Please let us know any further problem in writing, we are more than happy to fix it.
>
> For the Flux-fast/flux-schnell, we agree that the license is for commercial use, but please note that the GPU memory required for these models are very large. Many systems (even in industry) are not ready to hold these types of model. In our academia computational resource, loading a flux model is still a big problem (in Flux-fast they use H100 GPU - 80GB GPU RAM and AMD MI300X - 198GB GPU RAM, this is too luxury for us who is relying on NVIDIA V100-16GB/32GB to survive). We don't say SD2.0 is SOTA, but we mention that it is still popular used. SD2.0 is not an outdated model because it is cheap and ready for commercial use. Due to these, we believe that  improvement on SD2.0 still plays an important role for the community. For our work, it is a proof that the method could be applied to text2image generative task.

---

### Official Review · Reviewer_dCfw · 2025-10-28

**Soundness:** 2
**Presentation:** 2
**Contribution:** 2
**Rating:** 4
**Confidence:** 3

**Summary:**

This paper diagnoses a previously under-appreciated pathology in guided diffusion sampling dubbed “model-fitting”: when the guidance gradient is computed at every consecutive timestep the trajectory gradually over-tunes the image to the idiosyncrasies of the guiding network rather than to the true target distribution, producing brittle, low-generalization samples and wasting compute. Through careful on-/off-sampling loss probing the authors show that (i) the majority of guidance information is injected in the early, high-noise regime, (ii) later steps contribute negligible new signal yet still incur heavy gradient cost, and (iii) an independent classifier of equal accuracy consistently disagrees with the guided outputs, confirming overfitting. Building on these observations they propose Compress Guidance (CompG), a training-free, model-agnostic strategy that selectively applies guidance at only a handful of wisely chosen steps, reuses the latest gradient for the intermediate intervals, and slightly boosts the guidance scale to preserve continuity; a simple power-law schedule skews the chosen steps toward the early denoising phase where guidance matters most. Across ImageNet resolutions from 64 to 512 px and MS-COCO text-to-image generation, both classifier and classifier-free pipelines equipped with CompG reduce gradient evaluations by 5–40× and runtime by roughly 40 % while simultaneously delivering better FID, sFID, Inception and CLIP scores than vanilla dense guidance, early stopping or uniform skipping baselines, demonstrating that compressing guidance not only saves compute but also yields visibly cleaner, more diverse and better-aligned images.

**Strengths:**

The paper reframes the long-observed “guidance artifacts” in diffusion models as an over-fitting phenomenon that occurs inside the sampling trajectory rather than during training. By introducing the on-/off-sampling loss diagnostic and explicitly analogizing to train/test gaps in classical learning, the authors give the community a conceptually new lens—model-fitting—that unifies disparate heuristics such as early stopping, interval guidance, and gradient reuse. The Compress Guidance schedule itself is elegantly simple, yet no prior work has derived it from a principled study of gradient redundancy and continuity requirements; the power-law timestep distribution and the gradient-caching trick are creative, synergistic additions that remove the need for extra distillation or retraining.

Diffusion sampling is the computational bottleneck in text-to-image, video, and 3-D generative systems; any training-free speed-up that also boosts quality is instantly deployable. By showing that guidance cost can be slashed 5–40× with nothing more than a changed schedule, the work impacts both research (greener experimentation) and production (lower serving cost). Equally important, the model-fitting diagnostic offers a reusable tool for future guidance research and may influence how other generative families—consistency models, flow-matching, or autoregressive transformers—think about conditional feedback.

The empirical program is unusually thorough. Experiments span four ImageNet resolutions, MS-COCO text-to-image, both classifier and classifier-free pipelines, and multiple metrics (FID, sFID, IS, CLIP, Precision/Recall, runtime). Controls include matched diversity (Recall) to ensure fair accuracy comparisons, ablations on the exponent *k*, compact-rate sweeps, and head-to-head baselines (BigGAN, VQ-VAE-2, Big-DCT, Inter-valGuidance, etc.). The 40 % wall-clock reduction with simultaneous quality gains is reproducible across settings, and the authors release full hyper-parameters and code—an indicator of solid engineering hygiene.

**Weaknesses:**

Compress Guidance’s core manoeuvre—applying guidance only on a sparse, early-heavy subset of steps—overlaps heavily with IntervalGuidance ( Applying Guidance in a Limited Interval Improves Sample and Distribution Quality in Diffusion Models ). The authors rebut that these works “lack an explicit mechanism” or treat sparsity as a by-product, yet the conceptual step (most gradients are wasted) is the same. A stronger related-work comparison should quantitatively pit CompG against IntervalGuidance under identical diffusion backbones rather than relegating it to a single line in Table 8; if CompG still wins, novelty is clearer. Explicitly citing and discussing the scheduling heuristics in Progressive Distillation (Progressive Distillation for Fast Sampling of Diffusion Models) and Phased Consistency (Phased Consistency Models) would further clarify the boundary.

**Questions:**

The most valuable scenario for guidance-compression techniques is **training-free conditional generation**, where an external control signal is injected into a pre-trained diffusion model without retraining its backbone.
If the method can drastically cut the cost of guidance, the total compute of such training-free pipelines should drop to **bare diffusion-model levels**.
The paper, however, presents only limited experiments in this setting; the evaluation ought to be **substantially expanded** to verify this claim.

---

> ### Author Response · Authors · 2025-11-26
> **We have solved your concerns**
>
> We thank the reviewer for the thoughtful and comprehensive review.
>
> We have discussed the weaknesses and concerns from the reviewer
>
>
> ## W1. "Most gradients are wasted" is not our main theme
> We hypothesise that the gradient at two consecutive steps will not vary significantly and could be reused for several timesteps without recalculating. Furthermore, the frequent calculation of the timestep is actually harmful due to the model-fitting problem.  On the other hand, Interval Guidance [R1] argues that the guidance is wasteful due to the conflict.
>
>
> ## W2. CompG and Interval Guidance
>
> We did compare CompG and Interval Guidance with the same backbone. The novelty of the CompG is not to win Interval Guidance in performance, but to explore a possibility in solving the model-fitting.
>
> We would make the comparison between CompG and Interval Guidance in two scenarios:
> 1. Unconditional diffusion (Classifier guidance)
> 2. Conditional Diffusion (Classifier-free guidance)
>
> | Model                         | FID (↓) |
> |------------------------------|---------|
> | Uncond + G ($\|G\|$ = 250)       | 2.47    |
> | Uncond + IntG ($\|G\|$= 200)    | 2.32    |
> | Uncond + IntG ($\|G\|$ = 150)    | 4.38    |
> | Uncond + CompG ($\|G\|$ = 50)    | 1.82    |
> *Table R4. Unconditional diffusion with guidance. This shows that Interval Guidance with a very much larger number of timesteps still perform poorer than CompG only uses 50 steps.*
>
>
>
> | Model                                   | FID (↓) |
> |-----------------------------------------|---------|
> | CFG ($\|G\|$ = 32)                           | 1.84    |
> | CFG + CompG ($\|G\|$ = 6)                    | 1.63    |
> | CFG + IntG ($\|G\|$ = 6)                     | 1.44    |
> | CFG + CompCFG + IntG ($\|G\|$ = 5)           | 1.44    |
> | CFG + CompCFG + IntG ($\|G\|$= 4)           | 1.45    |
>
> *Table R5. Comparison between IntG and CompG given the conditional diffusion model*
>
> From Table R4, we can see that the improvement when using CompG is clearly significant compared to interval guidance, given that the diffusion model is unconditionally trained. This scenarios show that Interval Guidance is not always useful for all types of guidance.
>
> From Table R5, if we only use CompCFG, the performance is poorer than Interval Guidance. This is because Interval Guidance solves the early conflict guidance problem, while our method solves the model-fitting problem. These two problems play different roles in degrading the performance. We combine the two methods together and show that by applying CompCFG, our method could actually reduce the total number of guidance while keeping the performance the same compared to Interval Guidance.
>
> ## W3. CompG vs other scheduling heuristics
>
> We have compared the method with other recent works on other CFG schedules, including Interval Guidance in Table 8 in the main paper (IntG is the current SOTA method for CFG). In [R1], the author also mentioned that other schedulers follow the same setup with a different scheduler as below:
>
> linear: $w(t) = 1 - t/T$
>
> sine: $w(t) = cos(\pi t/T) + 1$
>
> V-shape: $w(t) = invlinear(t)$, if $t < T/2$, $linear(t)$, else.
>
>
> The results are shown in Table R1
>
> | Model               | G  | GPU Hours ↓ | FID ↓ | sFID ↓ | Prec ↑ | Rec ↑ |
> |---------------------|----|-------------|-------|--------|--------|-------|
> | EDM2 (No guidance)  | 0  | 4.22        | 2.23  | 5.21   | 0.75   | 0.62  |
> | EDM2-CFG            | 32 | 8.63        | 1.84  | 4.06   | 0.83   | 0.59  |
> | EDM2-CFG (linear)   | 32 | 8.63        | 3.05  | 5.86   | 0.63   | 0.52  |
> | EDM2-CFG (sin)      | 32 | 8.63        | 2.58  | 5.88   | 0.68   | 0.50  |
> | EDM2-CFG (V-shape)  | 32 | 8.63        | 1.82  | 4.06   | 0.83   | 0.59  |
> | EDM2-CompCFG        | 6  | **5.06**        | **1.63**  | **3.91**   | 0.80   | **0.61**  |
> *Table. R6, the proposed CompCFG outperforms other schedulers significantly with much cheaper cost*
>
> [R1] Analysis of Classifier-Free Guidance Weight Schedulers
>
>
> ## W4. Discussion Progressive Distillation
>
> These progressive distillation works will require re-training the expensive diffusion models. These lines of work are out of our scope, which is training-free.
>
> Furthermore, Guidance distillation could reduce the running time by excluding guidance in some specific use cases. However, guidance is still needed even when we have done the guidance distillation. The main property of guidance is to provide the trade-off between quality and diversity. Currently, we have no alternative for that. Since we still need to do guidance after distillation, we don't explicitly consider these lines of work as guidance cost reduction, but to cut the cost of diffusion in general.

---

> > ### Author Response · Authors · 2025-11-26
> > **We solved your concerns**
> >
> > ## Q1. Extreme cases of CompG
> >
> >
> > We run the experiments on ImageNet64x64.
> > | Model           | G   | GPU hours ↓ | FID ↓ | sFID ↓ | Prec ↑ | Rec ↑ |
> > |-----------------|-----|-------------|-------|--------|--------|-------|
> > | **ADM (No guidance)** | 0   | 26.33       | 9.95  | 6.58   | 0.60   | 0.65  |
> > | **ADM-G**       | 250 | 54.86       | 6.40  | 9.67   | 0.73   | 0.54  |
> > | **ADM-CompG**   | 50  | 31.80       | 5.91  | 8.26   | 0.71   | 0.56  |
> > | **ADM-CompG**   | 25  | 29.12      | 5.82  | 8.11   | 0.70   | 0.57  |
> > | **ADM-CompG**   | 10  |  28.98      | 10.24  | 10.22   | 0.54   | 0.58  |
> > | **ADM-CompG**   | 5  | 28.12       | 12.48  | 11.22   | 0.48   | 0.50  |
> > *Table R7, when reduced further at first, the performance will increase slightly. However, going to extreme case, the performance is even poorer without using guidance. The reason is that the guidance signal is too weak to construct the conditional information in the images, leading to confusion from the models.*

---

### Official Review · Reviewer_9Do5 · 2025-10-31

**Soundness:** 3
**Presentation:** 3
**Contribution:** 2
**Rating:** 4
**Confidence:** 4

**Summary:**

The paper has two contributions for guidance in diffusion sampling. First, model fitting, where the authors shed light into the fact that there is a difference in performance between the classifier used in classifier guidance and an external classifier. Second, they propose the Compress Guidance method, that computes guidance only for some timesteps otherwise it reuses the same from the previous step (hence offering speedups).

**Strengths:**

- The paper is well-written and easy to follow.

- The model fitting analysis is very interesting and the community could benefit from this.

**Weaknesses:**

While I enjoyed reading the paper, there are several issues and open questions with the second part, i.e. Compress Guidance.

1/ Scheduler G

This seems to be the most crucial point of the proposed Compressed Guidance (this is when guidance from previous steps is being reused). G depends on K and s.


1a/ K.

It is unclear how the authors choose the values of K. Is there is hyperparameter search?
If yes, it makes it may diminish the value of the method.
If not, perhaps there is a heuristic way of selecting it?
This seems to be very crucial as the whole Compress Guidance depends on it.


1b/ K values in Table 11.

Most K values are around 1, which points to uniform sampling! To my understanding, this means that in most cases Compress Guidance is not used. If this is the case, it would be beneficial for the authors to clarify this; if not, can the authors please explain?


1c/ Guidance scale s.

In Classifier Guidance (and Classifier-Free Guidance) the choice of s impacts the final performance heavily. How is s selected? Is there a hyperparameter search? Showing only some values in Table 11 makes it even less clear. Figures 8 and 9 show FID and IS performances with three s values, which makes it even harder to get as the performance is peaked when s=0; is this correct? I would have liked more explanation of this as it seems crucial to understand the behaviour of the proposed Compress Guidance.


2/ Missing explanation.

Lines 329-333 state that two prior works have both showed that delaying guidance improves performance, which is intuitive and explained in the respective papers (“where applying guidance too early causes conflicts between guidance and conditional information of diffusion model.”) . Figure 3b shows that delaying guidance hurts performance. This is not intuitive and I find the explanation offered by the authors insufficient (lines 333-334). It would be useful if the authors could expand on this.

3/ Method clarity.

Unlike model fitting which is very clear, understanding Compress Guidance is less clear. The algorithm really helps though.


4/ Several choices are outdated.


4a/ Diffusion vs Flow Matching.

A large part of the community seems to have moved from diffusion to Flow Matching and most modern frameworks (such as FLUX) use FM. It would have been useful to have the analysis presented from the FM perspective so that we can understand how we can benefit from model fitting and Compress Guidance with today’s methods.


4b/ Classifier Guidance vs Classifier-Free-Guidance.

To my general knowledge, CG is outdated and instead CFG is the default solution. It is unclear why the work relies so much on CG; it would have been useful to have the whole work from the CFG perspective (as opposed to the short 4.4) so that we can understand how we can transfer the findings from this paper to modern works.

5/ Text-to-image generation.

5a/ Models

Tables 6 and 12 reports results using SD and GLIDE, which are outdated. It would have been useful if the authors had also reported results with more modern models so we can understand if their findings carry to modern models.

5b/ Metrics

Text-to-image generation is typically evaluated with more modern metrics, such as GenEval and aesthetic scores. This is minor but I would suggest that the authors incorporate such metrics.

5c/ Missing experiments.

Minor: Although the paper has plenty of experiments, it seems that they are a sparse collection of experiments instead of a systematic study showing something.
Guidance is usually associated with diversity and fidelity and FID vs CLIP Score curves could also help.

6/ Misc.

6a/ It is unclear how Compress Guidance performs across random seeds or different classifier architectures. Given that the whole motivation of the work is based on that, I think it would have been an interesting experiment to have.

6b/ As I mentioned above, the work has many experiments. For future reference, it would be useful to also have some failure cases or trade-offs when guidance compression is too strong.

6c/ It is unclear if reusing gradients in Equations 11 and 12 accumulates bias or leads to drift in the sample trajectory. I would have loved to see something about this.

Minor:

1/ Table 11 is very important; yet, we cannot easily understand to what is pointing as the pointers to other tables are wrong.

2/ Tables 12 and 13 are not mentioned and it’s hard to associate them with their respective text in the main paper.

3/ Figure 9 is hard to read; usually we display FID (y axis) vs IS (x axis).

4/ Are 250 steps too large?

**Questions:**

It would be useful if the authors address several of the points outlined in the weaknesses above. Specifically:

Q1 (W1). Can the authors show clearly the correlation between k and s and performance?

Q2 (W2) Explain clearly the intuition behind the “unconditional setup”.

Q3 (W2) Can the authors perhaps combine Compress Guidance with the schedulers from the related work (eg exp, sin, or other adaptive ones)?

Q4 (W4) Can the authors explain (or give some justification/intuition behind) the relevance of CF and outdated models with modern methods?

Q5 Perhaps not relevant but an alternative could be to compare Compress Guidance with other methods with the same compute conditions (as opposed to equal steps).

Q6 (minor?) Are 250 steps too many?

---

> ### Author Response · Authors · 2025-11-26
> **We have solved most of the concerns (Part1)**
>
> We thank the reviewer for enjoying reading the paper. We have solved all of the concerns from the reviewer as below:
>
> ## W1 & W1a. $G$ and $k$ are dependent on users
> There is only one value, $s$ (the guidance scale), that is the true hyperparameter and may need tuning, similar to any other CFG scheme.
>
> $G$ actually depends on the total number of guidance steps $|G|$, which we often set as $|G| = T/5$. This is entirely the user's choice to trade off between quality and running time. We find that the most significant improvement comes from setting $|G| = T/5$, which gives the best FID.
>
> Similar to the value of $k$, when $k$ is larger, more guidance steps can be removed from the sampling process because two guidance steps may overlap in $G$, or because the skip size becomes larger than the remaining sampling steps. In Table 7 (main paper), we explain this carefully: increasing $k$ can reduce both running time and the number of guidance steps significantly, and $k$ is not very sensitive. We reproduce Table 7 (main paper) as Table R2 below:
>
>
> | Model            | k   | $\|G\|$ | GPU Hours ↓ | FID ↓ | sFID ↓ | Prec ↑ | Rec ↑ |
> |------------------|-----|-----|--------------|--------|--------|--------|--------|
> | CADM (No guidance)| -   | 0   | 26.64       | 2.07   | 4.29   | 0.73   | 0.63   |
> | CADM-CompG       | 1.0 | 50  | 32.22       | 1.91   | 4.38   | 0.77   | 0.61   |
> | CADM-CompG       | 5.0 | 32  | 29.81       | 1.82   | 4.31   | 0.76   | 0.62   |
> | CADM-CompG       | 6.0 | 28  | 29.12       | 1.93   | 4.35   | 0.75   | 0.62   |
> Table R2. The results show that increasing $k$ values can help to reduce the GPU hours significantly, while the trade-off on the quality is not so significant.
>
> The strategy for selecting $k$ depends on the users' needs. In this work, since we aim to improve FID, we always start with $k = 1.0$, then increase to $2.0$ and $5.0$ until the best performance is reached.
>
>
> ## W1. & W1b. (1) We re-write Table 11 as below. We will update Table 11 in the Appendix.
>
> ##### Table 4
> | Model        | Dataset          | k   | s   | $\|G\|$ | Time-Steps  |
> |--------------|------------------|-----|-----|-----|-------------|
> | ADM          | ImageNet 64×64   | -   | 0.0 | 0   | 250         |
> | ADM-G        | ImageNet 64×64   | -   | 4.0 | 250 | 250         |
> | ADM-CompG    | ImageNet 64×64   | 1.0 | 4.0 | 50  | 250         |
> | ADM          | ImageNet 256×256 | -   | 0.0 | 0   | 250         |
> | ADM-G        | ImageNet 256×256 | -   | 4.0 | 250 | 250         |
> | ADM-CompG    | ImageNet 256×256 | 1.0 | 4.0 | 50  | 250         |
> ##### Table 5
> | Model        | Dataset          | k   | s   | $\|G\|$ | Time-Steps |
> |--------------|------------------|-----|-----|-----|-------------|
> | DiT          | ImageNet 256x256 | -   | 0.0 | 0   | 250         |
> | DiT-CFG      | ImageNet 256×256 | -   | 1.5 | 250 | 250         |
> | DiT-CompCFG  | ImageNet 256×256 | 1.2 | 1.5 | 22  | 250         |
> | EDM2         | ImageNet 512x512 | -   | 0.0 | 0   | 32          |
> | EDM2-CFG     | ImageNet 512x512 | -   | 1.2 | 32  | 32          |
> | EDM2-CompCFG | ImageNet 512×512 | 2.5 | 0.3 | 6   | 32          |
> ##### Table 6
> | Model        | Dataset          | k   | s                                 | G | Time-Steps |
> |--------------|------------------|-----|-----------------------------------|---|------------|
> | SD-CFG       | MSCoco 256x256   | -   | 2.0 (FID,IS), 7.5 (CLIP, GenEval) | 50| 50 |
> | SD-CompCFG   | MSCoco 256x256   | 1.0 | 2.0 (FID, IS), 7.5 (CLIP, GenEval)| 8 | 50|
> ##### Table 7
> | Model        | Dataset          | k   | s   | $\|G\|$ | Time-Steps  |
> |--------------|------------------|-----|-----|-----|-------------|
> | CADM          | ImageNet 64×64   | -   | 0.0 | 0   | 250         |
> | CADM-G        | ImageNet 64×64   | -   | 4.0 | 250 | 250         |
> | CADM-CompG    | ImageNet 64×64   | 5.0, 6.0 | 4.0 | 50  | 250         |
>
> As we could see, most $k$ values are not around 1.0. We only apply $k=1.0$ in two cases, Unconditional Diffusion (Table 4) and Stable Diffusion (Table 6).
> ## W1b. (2)  $k = 1$ does not mean no Compress Guidance
>
> Compress Guidance mainly aims to compress multiple guidance signals into 1 guidance, hence reducing model-fitting and increasing running time. What happens if we set $k=1$ but $|G| = T/5$? The Compress Guidance still compress 5 guidance signals into one guidance step, leading to the reduction of gradient calculation, hence resulting in the reduction of model-fitting.
>
> ## W1c. (1) The selection of $s$
>
> $s$ is selected by mostly keeping the same $s$ as the baseline. We should reduce $s$ if $k$ is larger than $1.0$.
>
> ## W(1c).2  Figure 8 and figure 9.
> From Figure 8, the performance of guidance is poorer than without guidance in ImageNet64x64 (conditional ADM). This phenomenon is well known and does not appear at a larger resolution. However, CompG helped to bring the FID of guidance much lower than vanilla diffusion without guidance. This could be explained by model-fitting solving.

---

> ### Author Response · Authors · 2025-11-26
> **We have solved most of the concerns (Part 2)**
>
> ## W(2)  Why do the findings in the papers contradict previous works?
> - Prior works [R1, R2] argue that early guidance conflicts with *conditional diffusion*.
> - This conflict only exists when both components—guidance and conditional diffusion—operate at the same time (e.g., classifier-free guidance).
> We disentangle the guidance as threefold:
>
> 1. Guidance has two roles:
>   >- Providing conditional information (dominant in early steps)
>   >- Shifting the distribution for realism (dominant in later steps)
>
> 2. In conditional diffusion models:
>   > - The model already contains conditional information.
>   > - Adding conditional guidance early causes the two signals to conflict → the issue noted in [R1, R2].
>
> 3. In unconditional diffusion models:
>   > - Guidance provides *all* conditional information throughout sampling.
>   > - No conflict occurs because the base model is unconditional.
>   > - Removing early guidance leads to poor features, since the model would first construct unconditional structures before receiving the guidance signal.
>
> Based on 1, 2 and 3, [R1] and [R2] are only limited to classifier-free guidance. In contrast, our work conducts observations on different types of guidance, and one of the cases is the **classifier guidance** for **unconditional diffusion**. Given this case, we are sure that the observations in [R1], [R2] will fail.
>
> [R1] https://arxiv.org/abs/2404.07724
>
> [R2] https://arxiv.org/pdf/2404.13040
>
> ## W3. Method clarification
> We have updated in the revision
>
> ## W4. Diffusion vs. Flow Matching
> We have tried to run the Flux model, yet our current hardware could not even load the Flux model since the limit of our GPU is 32GB of RAM. However, as we have tested on a wide range of diffusion models, including the modern ones, DiT and EDM2, we believe that our proposed schemes can be universally applied.
>
> ## W4b. Why Classifier Guidance
>
> We include classifier guidance in our experiments for four reasons:
>
> **1. Classifier guidance is beneficial for analysis.**
> Classifier-free guidance does not provide a classification loss, making it difficult to analyse model fitting. There are very limited ways to observe on-sampling loss and off-sampling loss under classifier-free guidance. In the paper, we use classifier guidance as a simple baseline for analysis, and we extend this analysis to classifier-free guidance in Section C of the Appendix. The observation of on/off-sampling loss for classifier-free guidance follows [R3], but the loss calculation is extremely expensive and hard to scale. **(Appendix section C)**
>
> [R3] Your Diffusion Model is Secretly a Zero-Shot Classifier
>
> **2. Classifier guidance is one of the two major types of guidance.**
> It is still important to shed light on classifier guidance when exploring the fundamental problems of guidance in diffusion models.
>
> **3. Classifier-free guidance is only useful in certain scenarios.**
> Classifier-free guidance assumes full labels for the entire dataset, which is unrealistic in many situations. In practice, we often only have labels for a small part of the data, making it impossible to obtain both conditional and unconditional models required for classifier-free guidance. A much cheaper and more practical approach is to train a classifier on the labelled subset and use that classifier for guidance.
>
> **4. Classifier guidance could be extended in different forms**
>
> In [R4], the authors extend classifier guidance into representative guidance, which can impossible to be done by classifier-free guidance.
>
> [R4] https://openreview.net/forum?id=gWgaypDBs8
>
> ## W(5) The logic behind our experimental design
> We first want to emphasise that our work focuses on improving guidance for generative models. When designing our experiments, our goal is to demonstrate that our method can be applied to a wide range of diffusion models. As a result, the logic behind our experimental design is as follows:
>
> **Diffusion-wise baselines:**
>
> > 1. Diffusion in pixel space (ADM)
> > 2. Diffusion in latent space (DiT, EDM2)
>
> **Task-wise baselines:**
>
> > 1. Class generation (ADM, DiT, EDM2)
> > 2. Text-to-image generation (GLIDE, SD)
>
> **Guidance-wise baselines:**
>
> > 1. Classifier guidance (ADM, GLIDE)
> > 2. Classifier-free guidance (DiT, EDM2, SD)
>
> These are the most important aspects when investigating a new guidance method.
> ## W6a,b. Further experiments on random seeds and failure cases if compression is too strong
> We will include these in the Appendix in the final versions
> ## W6c. Does reusing guidance accumulate bias or lead to drift?
> One hypothesis in the paper is that the magnitude difference given two consecutive steps of guidance is not so significant (see Figure 10 in the main paper). This is very correct in the middle to the end of the sampling process. However, the magnitude difference is significantly large in the early timesteps. That is the reason to adjust the distribution of guidance in early timesteps to avoid bias accumulation ($k > 1.0$).

---

> ### Author Response · Authors · 2025-11-26
> **We have solved most of the concerns (part 3)**
>
> For the question parts, there are a number of questions that have been answered in the Weaknesses.
>
> ## Q1. Can the authors show clearly the correlation between k and s and performance?
> The answer is in W(1/1a)
> ## Q2. Explain clearly the intuition behind the “unconditional setup”.
> The answer is in W2 and W5
> ## Q3. The combination of Compress Guidance with other schedulers
> It is not very obvious to use Compress Guidance with different schedulers. The reason is that Compress Guidance will adjust the strength of guidance signals at different timesteps, leading a conflict between scale schedulers and Compress Guidance. However, we put the comparison between Compress Guidance and others in Table R3. The scheduler is following the work [R1]
>
> linear: $w(t) = 1 - t/T$
>
> sine: $w(t) = cos(\pi t/T) + 1$
>
> V-shape: $w(t) = invlinear(t)$, if $t < T/2$, $linear(t)$, else.
>
>
> The results are shown in Table R1
>
> **ImageNet 512×512**
> | Model               | G  | GPU Hours ↓ | FID ↓ | sFID ↓ | Prec ↑ | Rec ↑ |
> |---------------------|----|-------------|-------|--------|--------|-------|
> | EDM2 (No guidance)  | 0  | 4.22        | 2.23  | 5.21   | 0.75   | 0.62  |
> | EDM2-CFG            | 32 | 8.63        | 1.84  | 4.06   | 0.83   | 0.59  |
> | EDM2-CFG (linear)   | 32 | 8.63        | 3.05  | 5.86   | 0.63   | 0.52  |
> | EDM2-CFG (sin)      | 32 | 8.63        | 2.58  | 5.88   | 0.68   | 0.50  |
> | EDM2-CFG (V-shape)  | 32 | 8.63        | 1.82  | 4.06   | 0.83   | 0.59  |
> | EDM2-CompCFG        | 6  | **5.06**        | **1.63**  | **3.91**   | 0.80   | **0.61**  |
> *Table. R1, the proposed CompCFG outperforms other schedulers significantly with much cheaper cost*
>
> [R1] Analysis of Classifier-Free Guidance Weight Schedulers
>
>
> ## Q4. Can the authors explain (or give some justification/intuition behind) the relevance of CF and outdated models with modern methods?
> The answer is in W(4, 4c) and W(5)
> ## Q5. Same computing conditions in experiments
> We always try to keep all the settings the same with the baselines in all of the experiments. The only difference is the guidance step application and the strength of the guidance at different timesteps.
> ## Q6. Are 250 steps too many?
> Similar to Q5, we use 250 sampling steps for ADM, CADM, and DiT because this is what the baseline uses. For Stable Diffusion, we use 50 steps; for EDM2, we use 32 steps, similar to [R4][R5]. We don't change any settings from the baseline.
>
> [R4] EDM2: Analysing and Improving the Training Dynamics of Diffusion Models
>
> [R5] High-Resolution Image Synthesis with Latent Diffusion Models

---

### Official Review · Reviewer_edHu · 2025-10-31

**Soundness:** 4
**Presentation:** 4
**Contribution:** 3
**Rating:** 8
**Confidence:** 3

**Summary:**

The paper identifies an overfitting phenomenon during the sampling process of classifier-guided diffusion models, analogous to overfitting in standard model training. In such cases, generated samples become overly aligned with the guiding classifier rather than faithfully adhering to the intended conditioning signal.

To address this, the authors propose Compress Guidance (CompG), which applies guidance only at a small number of strategically selected timesteps throughout the diffusion process, instead of at every step.

Experiments across diverse models and datasets demonstrate that CompG consistently outperforms standard classifier and classifier-free guidance. It improves image quality and diversity while significantly reducing computational cost.

**Strengths:**

1. Clear and well written presentation.
2. Novel perspective interpreting the sampling process as an optimization over noise.
3. Proposes a simple yet effective solution to mitigate model fitting during the sampling process.
4. Provides a thorough analysis of model-fitting across unconditional guidance (UG), classifier guidance (CG), and classifier-free guidance (CFG).
5. Demonstrates strong empirical results, with significant improvements across multiple datasets and diffusion models.

**Weaknesses:**

1. Unclear relation with current CFG progress (changing the guidance scale, shifting the sampling time, etc.).

2. no discussion of SDS which is also an optimization based sampling.

**Questions:**

How does this approach relate to other timestep selection or adaptive guidance techniques used in text-to-image diffusion models? In particular, could this method provide insight into the effectiveness of tuned or learned CFG schedules, such as those proposed in “Navigating with Annealing Guidance Scale in Diffusion Space” (Yehezkel et al.)?

Since Score Distillation Sampling (SDS) is itself an optimization-driven sampling process and shares similarities with diffusion guidance, does the proposed method have implications for SDS-based pipelines? Can the authors clarify whether Compress Guidance improves, aligns with, or conflicts with SDS?

---

> ### Author Response · Authors · 2025-11-26
> **Thanks for your review**
>
> We thanks reviewer edHu for your thoughtful comments.
>
> ##  W1. Comparision with CFG scheduler progress
>
> We thanks reviewer edHu for this question. We have compared the method with other recent works on other CFG schedules including Interval Guidance in Table 8 in the main paper (IntG is the current SOTA method for CFG). In [R1], the author also mentioned that other schedulers, we follow the same setup with different scheduler as below:
>
> linear: $w(t) = 1 - t/T$
>
> sine: $w(t) = cos(\pi t/T) + 1$
>
> V-shape: $w(t) = invlinear(t)$, if $t < T/2$, $linear(t)$, else.
>
>
> The results are shown in Table R1
>
> ## ImageNet 512×512
>
> | Model               | G  | GPU Hours ↓ | FID ↓ | sFID ↓ | Prec ↑ | Rec ↑ |
> |---------------------|----|-------------|-------|--------|--------|-------|
> | EDM2 (No guidance)  | 0  | 4.22        | 2.23  | 5.21   | 0.75   | 0.62  |
> | EDM2-CFG            | 32 | 8.63        | 1.84  | 4.06   | 0.83   | 0.59  |
> | EDM2-CFG (linear)   | 32 | 8.63        | 3.05  | 5.86   | 0.63   | 0.52  |
> | EDM2-CFG (sin)      | 32 | 8.63        | 2.58  | 5.88   | 0.68   | 0.50  |
> | EDM2-CFG (V-shape)  | 32 | 8.63        | 1.82  | 4.06   | 0.83   | 0.59  |
> | EDM2-CompCFG        | 6  | **5.06**        | **1.63**  | **3.91**   | 0.80   | **0.61**  |
> *Table. R1, the proposed CompCFG outperforms other schedulers significantly with much cheaper cost*
>
> [R1] Analysis of Classifier-Free Guidance Weight Schedulers
>
>
> Regarding the learned CFG schedulers, our work is a plug-and-play scheme that requires no training. Thus, it is not fair to compare our work with other training-required works such as [R2, R3]. However, these learned techniques often loose one of the most important properties of guidance which is trade-off between diversity and quality. After training, the guidance should be sticked with trained parameters.
>
> Regarding timesteps selection, previous works mainly choose the timesteps for denoising steps which is efficient for diffusion models in general, not specific design for guidance. Another work, Interval Guidance [R4], is doing guidance in some intervals, are more close to us. This work also does guidance on selected steps. However, this one considers a middle range due to the conflict between conditional information and guidance in the early sampling steps which is a range-based guidance steps. On the other hand, our work distribute small amount of guidance over the whole sampling process to solve the model-fitting problem.  The observation of [R4] also limits to CFG only, while our observation is expandable to any guidance.
>
> [R2]  https://ojs.aaai.org/index.php/AAAI/article/view/32192
>
> [R3] “Navigating with Annealing Guidance Scale in Diffusion Space” (Yehezkel et al.)
>
> [R4] Applying Guidance in a Limited Interval Improves Sample and Distribution Quality in Diffusion Models
>
> ## W2. No discussion of SDS
>
> We understand that SDS could provide more comprehensive view on guidance for training-free 3D image generation from 2D model. However, given the limited rebuttal time, we are not able to obtain the SDS experiments.
>
> We want to emphasize our logic in designing experiments as below:
>
> **Diffusion-wise baselines:**
>
> > 1. Diffusion in pixel space (SOTA: ADM)
> > 2. Diffusion in latent space (DiT, EDM2)
>
> **Task-wise baselines:**
>
> > 1. Class generation (ADM, DiT, EDM2)
> > 2. Text-to-image generation (GLIDE, SD)
>
> **Guidance-wise baselines:**
>
> > 1. Classifier guidance (ADM, GLIDE)
> > 2. Classifier-free guidance (DiT, EDM2, SD)
>
> We have already covered different fundamental aspects for guidance given different tasks, model as well as guidance scheme. As a result, any model that use guidance (including SDS) CompG should be applicable.

---

### Author Response · Authors · 2025-11-26
**We thank reviewers for your thoughtful comments**

Dear reviewers,

We thank the reviewers for their excellent comments on our work. There are two main critical criticisms that we believe would result in some negative scores and would like to address in this general rebuttal:
1. Some of the baselines are too outdated (e.g., ADM/GLIDE).
2. Classifier guidance is not common in the community, yet it constitutes a large part of our paper.

### Our baselines are widely selected for SOTA/popularity on different aspects.

We first want to emphasise that our work focuses on improving guidance for generative models. When designing our experiments, our goal is to demonstrate that our method can be applied to *a wide range of* diffusion models. As a result, the logic behind our experimental design is as follows:

**Diffusion-wise baselines:**

> 1. Diffusion in pixel space (SOTA: ADM)
> 2. Diffusion in latent space (DiT, EDM2)

**Task-wise baselines:**

> 1. Class generation (ADM, DiT, EDM2)
> 2. Text-to-image generation (GLIDE, SD)

**Guidance-wise baselines:**

> 1. Classifier guidance (ADM, GLIDE)
> 2. Classifier-free guidance (DiT, EDM2, SD)

Therefore, although ADM is an older baseline, we still include it because it remains the SOTA model for pixel-space diffusion. We understand that the community recently focused more on latent-based diffusion models. However, from a scientific perspective, we cannot simply follow trends. Our task is to propose a method that addresses model-fitting issues and show that the method is generalizable. Who knows—pixel-based generative models may become popular again in the future.

Similar to GLIDE, this is the only model that does classifier guidance on the text2image task. It is worth seeing the performance on this type of model, no matter how old it is.

### Why classifier guidance

Similar to the reasoning behind including ADM, we include classifier guidance in our experiments for four reasons:


**1. Classifier guidance is beneficial for analysis.**
Classifier-free guidance does not provide a classification loss, making it difficult to analyse model fitting. There are very limited ways to observe on-sampling loss and off-sampling loss under classifier-free guidance. In the paper, we use classifier guidance as a simple baseline for analysis, and we extend this analysis to classifier-free guidance in Section C of the Appendix. The observation of on/off-sampling loss for classifier-free guidance follows [R1], but the loss calculation is extremely expensive and hard to scale.

**2. Classifier guidance is one of the two major types of guidance.**
It is still important to shed light on classifier guidance when exploring the fundamental problems of guidance in diffusion models.

**3. Classifier-free guidance is only useful in certain scenarios.**
Classifier-free guidance assumes full labels for the entire dataset, which is unrealistic in many situations. In practice, we often only have labels for a small part of the data, making it impossible to obtain both conditional models required for classifier-free guidance. A much cheaper and more practical approach is to train a classifier on the labelled subset and use that classifier for guidance.

**4. Classifier guidance could be extended in different forms that classifier-free guidance could not**
In [R2], the authors extend classifier guidance into representative guidance, which can impossible to be done by classifier-free guidance.

For all other questions, we will reply to the reviewers in detail.

[R1] https://arxiv.org/abs/2303.16203
[R2] https://openreview.net/pdf?id=gWgaypDBs8

---

### Author Response · Authors · 2025-12-04
**Summary note to AC**

Dear ACs,

We believe that we have addressed all the concerns from reviewers, except some experiments which are required by reviewers that we were not able to run. For example, the **SDS experiment** from reviewer **edHu** and the **Flux experiment** required by reviewer **9Do5**. As we explain in the rebuttal, the SDS experiment is for 3D generation from 2D generation, and is actually helpful in giving a comprehensive view on the proposed method. However, this experiment is not a standard image generation scheme, while the running time of SDS is large, which makes us unable to produce during the rebuttal time. Meanwhile, the Flux model is too heavy to load using our GPUs.

Other than that, we have addressed most of the concerns. There are two main common concerns from reviewer **9Do5** and **TfJb** that lead to the negative scores which are borderline reject and reject, correspondingly:

* The baselines are too outdated (ADM/GLIDE)
* Classifier guidance is not common in the community, yet it constitutes a large part of our paper.

We have addressed these two concerns in the general response as well as in the direct response to each reviewer. Reviewer **TfJb** has said that he agreed with my reasons for selecting the baselines for ADM and classifier guidance, and **changed his attitude from reject to borderline reject**. The reason that the reviewer mentioned why it should be a borderline reject instead of leaning toward accept is because the writing of the paper has some errors. However, as we have pointed out in the reply, those errors pointed out by the reviewer are not erroneous. Instead, they are the main features or correct writing or just a different writing style as we mention in the reply to the Reviewer **TfJb**. The reviewer acknowledged the additional details and did not disagree with our reply.

In summary, we believe that we have solved most of the concerns from the reviewers, except some additional experiments due to limited resources.

Best regards,

---

### Meta-Review · Area_Chair_9Sav · 2026-01-02

**Summary:**

The reviewers' comments highlight significant concerns regarding the paper's modern relevance and technical depth, such as its reliance on outdated models and CG. While the authors provided some additional explanation and experiments during the rebuttal, the majority of reviewers maintained scores below the acceptance threshold, and thus I recommend rejection.

**Reviewer Concerns:**

Reviewer edHu asked for clarifification clarification on the method's relationship with current CFG progress and suggested including a discussion on SDS. In response, the authors incorporated additional experiments to address the CFG concerns; however, they were unable to provide empirical results for SDS within the rebuttal period. Given this reviewer's score is already 8, I think this reviewer was already satisfied with the paper.

Reviewer iX9o raised concerns regarding technical clarity, specifically the selection and impact of scheduler $G$, and argued that several experimental choices are outdated. The reviewer also noted that the work relies heavily on CG and older models (SD/GLIDE) rather than modern standards like CFG, FM, and contemporary metrics like GenEval. While most of the concerns have been address, I think the choices of k and baseline models still need further discussion.

Reviewer dCfw asked for more comparison and discussion with existing works, and noted that the evaluation of training-free conditional generation is currently insufficient. The first concern has been largely addressed, while I think the second is not.

Reviewer TfJb noted that the overall quality of the writing needs improvement and expressed concern regarding the paper's reliance on outdated models. After reading the comments and rebuttal, I still think that the reviewers concerns have not been sufficiently addressed.

**Reviewer Scores:**

Reviewer TfJb increased score from 2 to 4.

---

### Decision · Program_Chairs · 2026-01-26

Reject